

# Automatic, high-order, and adaptive algorithms for Brillouin zone integration

**Jason Kaye$^{1,2\star}$, Sophie Beck$^{1\dagger}$, Alex Barnett$^2$,**
**Lorenzo Van Muñoz$^3$ and Olivier Parcollet$^{1,4}$**

**1** Center for Computational Quantum Physics, Flatiron Institute,
162 5th Avenue, New York, NY 10010, USA
**2** Center for Computational Mathematics, Flatiron Institute,
162 5th Avenue, New York, NY 10010, USA
**3** Department of Physics, Massachusetts Institute of Technology,
77 Massachusetts Avenue, Cambridge, MA 02139, USA
**4** Université Paris-Saclay, CNRS, CEA, Institut de Physique Théorique,
91191, Gif-sur-Yvette, France

$\star$ jkaye@flatironinstitute.org , $\dagger$ sbeck@flatironinstitute.org

## Abstract

We present efficient methods for Brillouin zone integration with a non-zero but possibly very small broadening factor $\eta$, focusing on cases in which downfolded Hamiltonians can be evaluated efficiently using Wannier interpolation. We describe robust, high-order accurate algorithms automating convergence to a user-specified error tolerance $\varepsilon$, emphasizing an efficient computational scaling with respect to $\eta$. After analyzing the standard equispaced integration method, applicable in the case of large broadening, we describe a simple iterated adaptive integration algorithm effective in the small $\eta$ regime. Its computational cost scales as $\mathcal{O}(\log^3(\eta^{-1}))$ as $\eta \to 0^+$ in three dimensions, as opposed to $\mathcal{O}(\eta^{-3})$ for equispaced integration. We argue that, by contrast, tree-based adaptive integration methods scale only as $\mathcal{O}(\log(\eta^{-1})/\eta^2)$ for typical Brillouin zone integrals. In addition to its favorable scaling, the iterated adaptive algorithm is straightforward to implement, particularly for integration on the irreducible Brillouin zone, for which it avoids the tetrahedral meshes required for tree-based schemes. We illustrate the algorithms by calculating the spectral function of $SrVO_3$ with broadening on the meV scale.



# 1  Introduction

Brillouin zone (BZ) integration is a fundamental operation in electronic structure calculations for periodic solids, and requires algorithms capable of accurately computing observables of physical systems with widely differing properties [1]. In some cases, BZ integration arises as a post-processing step, for example in the calculation of the spectral function and optical response functions. In others, it represents a fundamental step of an iteration procedure, and must be carried out repeatedly with controlled accuracy. Examples of the latter include the self-consistent field calculation of ground state properties within density functional theory [2], and the self-consistency loop in dynamical mean-field theory (DMFT) [3]. In both cases, BZ integration can represent a computational bottleneck or a significant source of error, particularly when high resolution is required to resolve fine features in reciprocal space.

In this article, we focus on many-body Green's function methods, in which the BZ integral acquires a system- and temperature-dependent scattering rate which largely determines the difficulty of integration. Although the ideas discussed here may be applied to other types of BZ integrals, we use the single particle retarded Green's function as a concrete prototypical example. It is given by

$$G(\omega) = \int_{\mathrm{BZ}} d^d\boldsymbol{k} \, \mathrm{Tr}\left[(\omega + \mu - H(\boldsymbol{k}) - \Sigma(\omega))^{-1}\right], \tag{1}$$

where BZ denotes the Brillouin zone, $\boldsymbol{k}$ is a reciprocal space vector, $\omega$ is a frequency variable (we take $\hbar = 1$), $\mu$ is the chemical potential, $H(\boldsymbol{k})$ is a Hermitian Hamiltonian matrix, $\Sigma(\omega)$

Figure 1: The first three panels show the energy sheets $\epsilon_1$, $\epsilon_2$, and $\epsilon_3$ (eigenvalues of $H(\boldsymbol{k})$) for the SrVO$_3$ example discussed in Section 6, with fixed $k_z = 1.9$. A plane of constant $\omega = 0.55$ eV is also shown. The last panel shows a color plot of the imaginary part of the integrand in (3) with this $\omega$, and $\eta = 0.01$ eV. The integrand concentrates in a region of width $\mathcal{O}(\eta)$ along the singular set defined by $\det\left(\omega - H(k_x, k_y, 1.9)\right) = \prod_{i=1}^{3}\left(\omega - \epsilon_i(k_x, k_y, 1.9)\right) = 0$, i.e., the intersections of the plane of constant $\omega$ with the three sheets.

is a complex-valued self-energy matrix, the dimension $d \in \{1, 2, 3\}$, and scalars are understood to be multiples of the identity matrix. Although in general $\Sigma = \Sigma(\boldsymbol{k}, \omega)$, we assume here that the self-energy is local in space, $\Sigma = \Sigma(\omega)$, a situation encountered in a variety of applications, in particular DMFT. The generalization to the case of non-local self-energies is problem-dependent and outside the scope of this work. The $\boldsymbol{k}$-integrated spectral function is obtained from (1) as

$$A(\omega) = -\frac{1}{\pi}\operatorname{Im} G(\omega). \tag{2}$$

To simplify the discussion we work with a special case of (1), with the self-energy $\Sigma(\omega) = -i\eta$ representing a constant scattering rate:

$$G(\omega) = \int_{\text{BZ}} d^d\boldsymbol{k}\,\operatorname{Tr}\left[(\omega - H(\boldsymbol{k}) + i\eta)^{-1}\right]. \tag{3}$$

Here, $\mu$ has been absorbed into $H(\boldsymbol{k})$. We emphasize, however, that our algorithms can be straightforwardly applied to the more general case.

In much of the literature, quantities of interest are considered in the limit of zero scattering (i.e., the non-interacting density of states is obtained from the $\eta = 0^+$ limit of (3)). In the presence of a self-energy, the scattering rate $\eta$ is non-zero, but may be small. For example, a local Fermi liquid has the asymptotic scaling [4]

$$\operatorname{Im}\Sigma(\omega, T) \sim -\left[\omega^2 + (\pi k_{\text{B}} T)^2\right], \tag{4}$$

as $\omega$, $T \to 0$, with $T$ the temperature, yielding a vanishing but non-zero scattering rate. In this case we obtain integrands in (3) with highly localized features at the scale $\mathcal{O}(\eta)$ near the surface on which $\det(\omega - H(\boldsymbol{k})) = 0$ (e.g. the Fermi surface when $\omega = 0$), as illustrated in Fig. 1. In order to compute (3) accurately, these localized features must be resolved, and adaptive methods become essential. In this article we present automatic algorithms to compute BZ integrals like (3), with possibly very small $\eta > 0$, to a user-specified error tolerance $\varepsilon$, with a cost scaling mildly with respect to $\eta$ and $\varepsilon$. Mild scaling with respect to $\varepsilon$ is achieved through the use of high-order accurate methods. The term "automatic" specifies that convergence to error $\varepsilon$ is carried out by the algorithm itself, not by the user. Indeed, within self-consistency loops, it is crucial that accurate results are returned reliably without user intervention.

An important distinction in BZ integration algorithms concerns the representation of $H(\boldsymbol{k})$. In many practical applications, the relevant physics can be extracted from a projection of the

ab initio Hamiltonian onto a low-energy subspace—a process referred to as downfolding—so that $H(\boldsymbol{k})$ in (3) becomes a small matrix. $H(\boldsymbol{k})$ is then related to a tight-binding Hamiltonian $H_{\boldsymbol{R}}$ by the Fourier series

$$H(\boldsymbol{k}) = \frac{1}{(2\pi)^d} \sum_{\boldsymbol{R}} e^{i\boldsymbol{k}\cdot\boldsymbol{R}} H_{\boldsymbol{R}}. \tag{5}$$

Here, $\boldsymbol{R}$ labels the real space lattice vectors of a tight-binding model, and mathematically the $H_{\boldsymbol{R}}$ are simply the matrix-valued Fourier coefficients of the periodic function $H(\boldsymbol{k})$. The gauge freedom in the Hamiltonian can be chosen, using (maximally) localized Wannier functions, to make $H_{\boldsymbol{R}}$ decay as rapidly as possible [5–7]. $H(\boldsymbol{k})$ can then be approximated efficiently using a truncation of (5). This procedure is typically referred to as Wannier interpolation [8]. We will not consider cases in which $H(\boldsymbol{k})$ is a large matrix, or cases in which Wannier interpolation is not applicable, as other algorithmic considerations then take precedence.

The literature on BZ integration is extensive, and we mention a few related approaches here. A standard method is to sum over a grid of equispaced nodes (such as the Monkhorst–Pack grid [9]), which we refer to as the $d$-dimensional periodic trapezoidal rule (PTR). Despite its spectral accuracy, to be discussed shortly, its obvious drawback is its inefficiency for small broadening $\eta$. A popular approach for the $\eta = 0^+$ limit of (3) is the linear tetrahedron method (LTM) [10–14], which uses a tetrahedralization of the BZ with piecewise linear approximation of band surfaces to obtain a semi-analytical result. There are relatively few works describing the extension of the method to $\eta > 0$, or more generally to (1) [15]. The LTM is low-order accurate, and further research is necessary to clarify its robustness, and its scaling with $\eta$. We give a brief discussion of the challenges and open questions associated with the LTM in Appendix E. We also mention smearing methods [16,17], used for the $\eta = 0^+$ case, which avoid computing distributional integrals appearing in the $\eta = 0^+$ limit by effectively adding a small broadening $\eta$ and then applying a uniform integration scheme. Adaptive smearing methods involve specific prescriptions for possibly $\boldsymbol{k}$-dependent broadening parameters [8, 18]; note that this is distinct from adaptive integration in the sense used in this article. Several other integration schemes have been proposed recently [19–21], but do not primarily address the class of problems considered here.

A few low-order accurate adaptive integration algorithms have been proposed. Refs. [22] and [23] describe tree-based adaptive integration methods, which are discussed below. Refs. [22] and [24] describe iterated adaptive integration algorithms which are close to that proposed here, but they have not gained widespread adoption. Our high-order accurate version of this approach, which includes critical performance optimizations, significantly improves its competitiveness.

The main contributions of this work are as follows:

- We present a simple, high-order accurate *iterated* adaptive integration (IAI) method, relying only on the use of a one-dimensional (1D) adaptive integration algorithm. It has $\mathcal{O}(\log^d(1/\eta))$ computational complexity as $\eta \to 0^+$. This is to be contrasted with the $\mathcal{O}(\eta^{-d})$ scaling of uniform integration methods like the PTR, which require an $\mathcal{O}(\eta)$ grid spacing to resolve features of width $\eta$.

- We observe that *tree-based* adaptive integration (TAI), another common approach [22, 23], scales only as $\mathcal{O}(\log(\eta^{-1})/\eta^{d-1})$, and is therefore asymptotically slower than IAI for small $\eta$. An example of the difference between the grids produced by TAI and IAI is given in Fig. 2 for the simple 2D example $H(k_x, k_y) = \cos(k_x) + \cos(k_y)$. Whereas TAI requires building adaptive tetrahedral meshes for calculations on the irreducible Brillouin zone (IBZ), IAI can be used directly on the IBZ with minimal modification.

- We include an in-depth discussion of the PTR, emphasizing its high-order accuracy. Despite its $\mathcal{O}(\eta^{-d})$ scaling, the PTR has significant advantages in the large $\eta$ regime. We

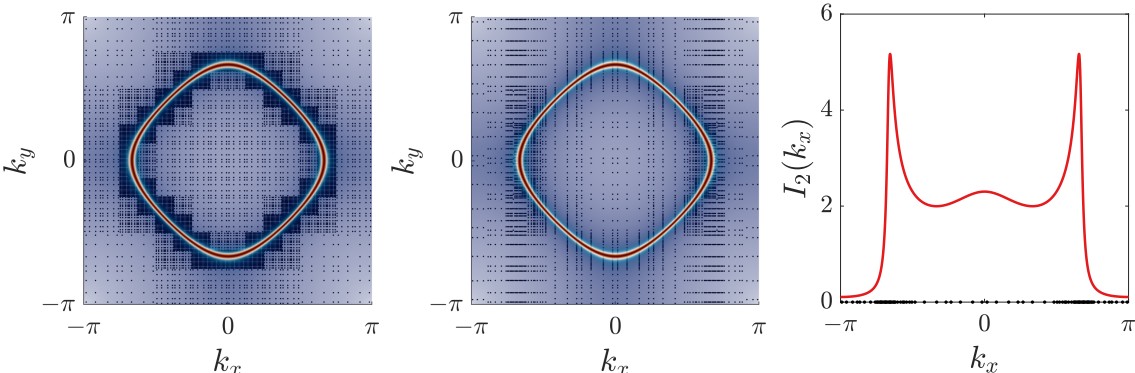

Figure 2: Adaptive integration grids for the spectral function (2) with $G$ given by (3) for $d = 2$, $H(k_x, k_y) = \cos(k_x) + \cos(k_y)$, $\omega = 0.5$ eV, $\eta = 0.05$ eV, and specified error tolerance $\varepsilon = 10^{-3}$ (in units of the spectral function). The TAI grid (left) refines along the full nearly-singular curve, yielding $\mathcal{O}(\log(\eta^{-1})/\eta)$ points (or $\mathcal{O}(\log(\eta^{-1})/\eta^2)$ in 3D). The integrand is shown as a color plot underneath the grid. By contrast, the IAI grid (middle) only refines into two nearly-singular points in $k_x$ (shown in the right panel), and then for each $k_x$, into at most two nearly-singular points in $k_y$, yielding $\mathcal{O}(\log^2(\eta^{-1}))$ points in total (or $\mathcal{O}(\log^3(\eta^{-1}))$ in 3D). The right panel shows the outer integrand $I_2(k_x) = -\frac{1}{\pi} \text{Im} \int_{-\pi}^{\pi} dk_y \left( \omega - H(k_x, k_y) + i\eta \right)^{-1}$, with its adaptive integration grid in $k_x$ shown as black dots.

show how the PTR can be used in an automated manner, and conclude that a hybrid approach, using the PTR for large $\eta$ and IAI for small $\eta$, provides an efficient and robust solution covering the full range of cases.

In addition, we present a high-order adaptive frequency interpolation method to represent the spectral function. This algorithm automatically generates an efficient grid to resolve localized features like Van Hove singularities, thereby minimizing the number of BZ integrals which need to be computed to obtain $A(\omega)$. We demonstrate our method by calculating the spectral function of $SrVO_3$ with constant scattering rates $\eta$ as small as 1 meV to several digits of accuracy.

This article is organized as follows. We begin in Section 2 with a discussion of the PTR and its advantages when $\eta$ is not too small. We review high-order 1D adaptive integration in Section 3, and compare the IAI and TAI approaches to higher-dimensional integration in Section 4, concluding that IAI is superior for BZ integration. We describe our automatic adaptive frequency sampling method in Section 5, and demonstrate the performance of the full scheme for a calculation of the spectral function of $SrVO_3$ in Section 6. Application areas and future directions of research are discussed in Section 7. We also include several appendices: Appendix A presents analysis of the PTR for BZ integrals, Appendix B gives details on the use of the PTR for integration in the IBZ, Appendix C discusses the efficient implementation of the PTR, Appendix D gives an alternative algorithm for automatic refinement of the PTR to that presented in the main text, and Appendix E discusses the LTM.

## Terminology and notation

We briefly fix some standard notation and terminology, and refer to Ref. [25, Chap. 3] for useful background. For concreteness, we restrict here to the 3D case.

The reciprocal lattice vectors $\boldsymbol{b}_1, \boldsymbol{b}_2, \boldsymbol{b}_3 \in \mathbb{R}^3$ are related to the primitive lattice vectors $\boldsymbol{a}_1, \boldsymbol{a}_2, \boldsymbol{a}_3 \in \mathbb{R}^3$ by $\boldsymbol{a}_i \cdot \boldsymbol{b}_j = 2\pi \delta_{ij}$, or equivalently, writing $A = \begin{bmatrix} \boldsymbol{a}_1 & \boldsymbol{a}_2 & \boldsymbol{a}_3 \end{bmatrix}$ and

$B = \begin{bmatrix} b_1 & b_2 & b_3 \end{bmatrix}$, by $A^T B = 2\pi I$, with $I$ the identity matrix. All $k$-dependent quantities, like $H(k)$, are periodic with respect to translation by the reciprocal lattice vectors, i.e., $H(k + Bn) = H(k)$ with $n \in \mathbb{Z}^3$. The term "Brillouin zone" often refers to the first Brillouin zone, the cell in the Voronoi decomposition of the reciprocal lattice containing the origin. However, this domain is in general a complicated polytope, and therefore less convenient for integration than the parallelpiped spanned by the reciprocal lattice vectors. Using the freedom afforded to us by the periodicity, we take this as our primitive unit cell, and refer to it as the Brillouin zone:

$$\text{BZ} \equiv \left\{ k = B\kappa/2\pi \,|\, \kappa \in [-\pi, \pi]^3 \right\}.$$

By changing to the $\kappa$ variables, we see that up to multiplication by a Jacobian factor $|\det B|/(2\pi)^3$ we can assume without loss of generality that $\text{BZ} = [-\pi, \pi]^3$, or equivalently, that $A = I$, $B = 2\pi I$, and $k = \kappa$. For simplicity of exposition, we make this assumption in the remainder of the article, except in discussions of IBZ integration, for which the specific point symmetries determine the integration domain.

The Fourier series representation (5) will be used to evaluate $H(k)$ efficiently. Under the assumptions made above, $R$ is an integer multi-index, $R = (R_1, R_2, R_3) \in \mathbb{Z}^3$. Truncating the rapidly converging Fourier series to include $M$ modes per dimension, we obtain

$$H(k) \approx \frac{1}{(2\pi)^3} \sum_{R_1, R_2, R_3 = -M/2}^{M/2 - 1} e^{ik \cdot R} H_R, \qquad (6)$$

where for simplicity we have assumed $M$ is even.

The spectral function has units of $(\text{eV\AA}^3)^{-1}$, but we suppress them throughout the text.

## 2 Periodic trapezoidal rule

As a starting point, we consider (3) with $\eta$ not too small. Although this case is less challenging than that of small $\eta$, the two scenarios are often encountered side by side in practice, for example in the presence of a self-energy $\Sigma(\omega)$ with $|\text{Im}\,\Sigma(\omega)|$ large for some $\omega$ and small for others. Taking the BZ as specified above, (3) is the integral of a smooth, triply-periodic function over a cube.

The standard tool in this case is the PTR. For a $2\pi$-periodic function $f(x)$ of one variable, this is simply the quadrature rule $\int_0^{2\pi} f(x)\,dx \approx \frac{2\pi}{N} \sum_{n=0}^{N-1} f(2\pi n/N)$, where we have shifted to the interval $[0, 2\pi]$ for notational convenience. We emphasize that although the ordinary trapezoidal rule for non-periodic functions is only second-order accurate, the PTR is *spectrally* accurate. In particular, the following theorem describes the error for analytic functions [26, Thm 3.2] (see also [27]).

**Theorem 1.** *If $f(x)$ is $2\pi$-periodic and analytic in the the strip $|\text{Im}\,x| < a$, with $|f(x)| \leq C$, then for any $N \geq 1$,*

$$\left| \int_0^{2\pi} f(x)\,dx - \frac{2\pi}{N} \sum_{n=0}^{N-1} f(2\pi n/N) \right| \leq \frac{4\pi C}{e^{aN} - 1}.$$

Thus the exponential rate of convergence is given by the distance from the real axis to the closest singularity of $f$ in the complex plane. As an example, in Fig. 3, we plot the error of the PTR against $N$ for the integral $\int_0^{2\pi} \frac{dk}{\sin(k) + i\eta}$, a 1D case of (3), for several choices of $\eta$. For large $\eta$, we observe exceptionally rapid convergence, yielding high accuracy with a coarse discretization, but for small $\eta$ localized features emerge and the convergence is slow. Indeed, Theorem 1 predicts error scaling close to $\mathcal{O}(e^{-\eta N})$, since a first-order Taylor expansion implies

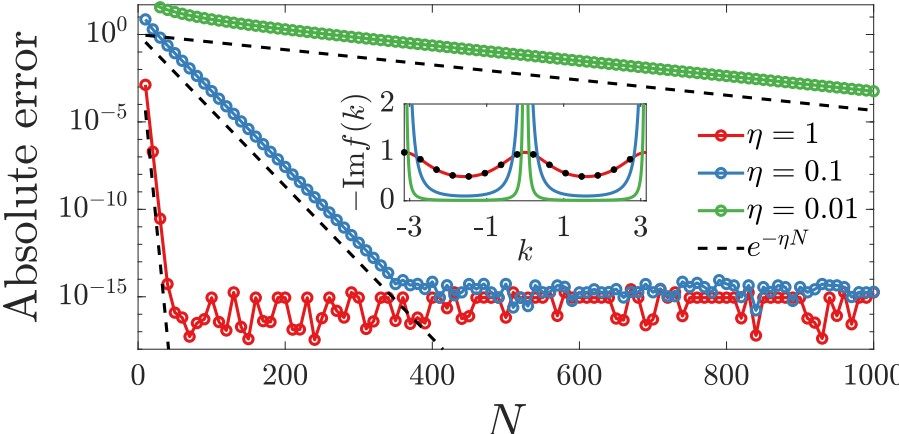

Figure 3: Error of the $N$-point PTR applied to $\int_{-\pi}^{\pi} \frac{dk}{\sin(k)+i\eta}$ with three values of $\eta$, along with a simple estimate of the convergence rate derived from Theorem 1. The negative imaginary part of the integrand $f(k)$ is shown in the inset. For $\eta = 1$, a relatively coarse discretization of $N = 15$ nodes (black dots) yields five-digit accuracy, and $N = 40$ yields full double-precision accuracy.

poles near $k = -i\eta, \pi + i\eta$. Appendix A makes this intuition rigorous for certain integrals of the form (3).

For $d > 1$, we can simply apply the PTR dimension by dimension; for example, in 2D,

$$\int_{[0,2\pi]^2} f(x,y)\, dx\, dy \approx \frac{(2\pi)^2}{N_1 N_2} \sum_{n_1,n_2=0}^{N_1-1,N_2-1} f\left( \frac{2\pi n_1}{N_1}, \frac{2\pi n_2}{N_2} \right). \tag{7}$$

The analogue of Theorem 1 for $d > 1$ is discussed in Appendix A. We again obtain exponential convergence, with rate determined by the height of the strip of analyticity in each variable, minimized over all other variables.

## 2.1 Periodic trapezoidal rule for Brillouin zone integrals

The PTR is therefore highly efficient for integrals of the form (3) when $\eta$ is sufficiently large. First, we can obtain the coefficients $H_{\boldsymbol{R}}$ in (6) by Fourier interpolation from samples $H(q_m)$ on a coarse, uniform grid of $M^d$ points $q_m$. The values $H(q_m)$ are typically obtained from an ab initio calculation of the band eigenvalues $\epsilon_n(k)$, transformed into the optimized Wannier basis. Then, using (6), $H(\boldsymbol{k})$ can be interpolated to a uniform integration grid of $N^d$ points $k_n$ and stored. Typically $H(\boldsymbol{k})$ is much smoother than the integrand in (3), so $N \gg M$, i.e., the grid required to accurately represent $H(\boldsymbol{k})$ can be made much coarser than the BZ integration grid. Finally, for each $\omega$, the PTR can be applied to (3), requiring the calculation of a matrix inverse at each of $N^d$ grid points.

We emphasize that $H(\boldsymbol{k})$ need only be computed once on the integration grid, stored, and reused for each $\omega$. Since the cost of evaluating $H(\boldsymbol{k})$ using (6) may be comparable to or greater than that of computing a small matrix inverse, this represents an advantage of the PTR over adaptive schemes. Indeed, any adaptive method must evaluate $H(\boldsymbol{k})$ on the fly for each new $\omega$, since the structure of an adaptive grid depends on the level set $\det(\omega - H(\boldsymbol{k})) = 0$ (see Fig. 1). The efficient evaluation of (6) for the PTR is discussed in Appendix C.

Although the IBZ is in general a complicated polytope (and in particular not a periodic domain), it can be shown that the PTR with $N_1 = N_2 = \cdots$ can be exactly *symmetrized* to a sum only over grid points within the IBZ, while maintaining spectral accuracy. Appendix B discusses this point in detail.

---

**Algorithm 1** Automatic PTR to compute $G(\omega)$.

---

**Inputs:** $\eta > 0$, $\omega$, $\varepsilon > 0$, positive integers $N_{\text{trial}}$ and $\Delta N$, arrays $H_{\text{trial}}$ and $H_{\text{test}}$ of values of $H(\boldsymbol{k})$ on PTR grids of $N = N_{\text{trial}}$ and $N = N_{\text{test}} \equiv N_{\text{trial}} + \Delta N$ points per dimension, $H_{\boldsymbol{R}}$.

**Output:** $G(\omega)$ correct to $\varepsilon$ accuracy, updated integers $N_{\text{trial}}$, $N_{\text{test}}$ and corresponding arrays $H_{\text{trial}}$, $H_{\text{test}}$.

    Using $H_{\text{trial}}$, $H_{\text{test}}$, approximate $G(\omega)$ by $N_{\text{trial}}^d$, $N_{\text{test}}^d$-
    point PTR to obtain $G_{\text{trial}}$, $G_{\text{test}}$
    $e \leftarrow |G_{\text{trial}} - G_{\text{test}}|$
    **while** $e > \varepsilon$ **do**
        $\{N, H, G\}_{\text{trial}} \leftarrow \{N, H, G\}_{\text{test}}$
        $N_{\text{test}} \leftarrow N_{\text{trial}} + \Delta N$
        Evaluate $H(\boldsymbol{k})$ on $N_{\text{test}}^d$ points to obtain $H_{\text{test}}$
        Compute $G(\omega)$ by $N_{\text{test}}^d$-point PTR to obtain $G_{\text{test}}$
        $e \leftarrow |G_{\text{trial}} - G_{\text{test}}|$
    **end while**
    $G(\omega) \leftarrow G_{\text{test}}$

---

## 2.2 Automatic integration with the periodic trapezoidal rule

The PTR can be used in a black-box algorithm to compute $G(\omega)$ in (3) to a user-specified error tolerance $\varepsilon$.

This algorithm estimates the error using a self-consistency check between the results calculated using the PTR with $N_{\text{trial}}$ and $N_{\text{test}} = N_{\text{trial}} + \Delta N$ points, respectively. Although it is stated for a single evaluation point $\omega$, in practice, given a collection of frequency points $\omega$, we repeatedly apply Algorithm 1, using the output values $\{N, H\}_{\text{trial}}$ and $\{N, H\}_{\text{test}}$ from the previous frequency point as inputs for the next one. In this way, as $G(\omega)$ is computed for each new $\omega$, the number of grid points is increased if necessary to ensure self-consistency to within $\varepsilon$. Since evaluating $H(\boldsymbol{k})$ is typically a computational bottleneck, reusing the arrays $H_{\text{trial}}$ and $H_{\text{test}}$ corresponding to the largest values of $N_{\text{trial}}$ and $N_{\text{test}}$ used so far is typically efficient, even if a less dense grid is required for good accuracy for a specific value of $\omega$. A refinement of the algorithm might maintain values of $H(\boldsymbol{k})$ on a collection of grids with different numbers of points, and for each new $\omega$ attempt to use the appropriate grid first. However, this is unlikely to provide a significant improvement as long as the cost of evaluating $H(\boldsymbol{k})$ is dominant. Most importantly, Algorithm 1 ensures that each value $G(\omega)$ is computed using a grid sufficiently dense to yield error below $\varepsilon$.

We remark that if $\Delta N$ is chosen too small, it is possible for the self-consistency check to provide an error estimate smaller than the true error, yielding a result with error greater than $\varepsilon$. The selection of $\Delta N$ depending on $\eta$ and the properties of $H$ is discussed in Appendix A. We also note that to avoid taking many refinement steps in each call of Algorithm 1, the initial value of $N_{\text{trial}}$ should be chosen proportional to $\eta^{-1}$. For the examples in Section 6, we take $N_{\text{trial}} = 6/\eta$.

We note that the procedure we have described allows the evaluation frequencies $\omega$ to be determined on-the-fly, between subsequent calls to Algorithm 1. This is required for the adaptive frequency interpolation algorithm described in 5, in which the evaluation frequencies are selected to discover and resolve localized features in the spectral function. In Appendix D, we suggest an alternative algorithm which may be advantageous in the alternative case that $G(\omega)$ is to be evaluated at a given, fixed set of frequencies.

# 3 Adaptive integration preliminaries

For sufficiently small $\eta$, resolving (3) using the PTR is infeasible. We saw a 1D example of this in Fig. 3, and refer again to Fig. 1 for an example of a highly localized integrand in the 2D case. Adaptive integration enables the efficient and automatic evaluation of singular or nearly-singular integrals with high-order accuracy. Before discussing higher-dimensional adaptive integration methods in Section 4, we first review Gauss quadrature for smooth non-periodic integrals in Section 3.1, and its use in 1D high-order adaptive integration in Section 3.2.

## 3.1 Gauss quadrature

The Gauss quadrature rule is given by

$$\int_a^b f(x)\,dx \approx \sum_{j=1}^p f(x_j)w_j\,,$$

where $x_j$ and $w_j$ are the nodes and weights of the rule, respectively. The rapid convergence properties of Gauss quadrature make it a standard tool for the integration of smooth, non-periodic functions. In particular, it is the unique $p$-node quadrature rule which integrates polynomials up to degree $2p-1$ exactly. The following theorem describes the error for analytic functions [28, Thm 4.5], [29], and should be compared with Theorem 1. We state the result for the standard interval $a = -1$, $b = 1$.

**Theorem 2.** *If $f$ is analytic in the region bounded by the ellipse with foci $\pm 1$ and major and minor semiaxis lengths summing to $\rho > 1$, with $|f(z)| \le C$, then for any $p \ge 1$,*

$$\left| \int_{-1}^1 f(x)\,dx - \sum_{j=1}^p f(x_j)w_j \right| \le \frac{64C}{15(1-\rho^{-2})\rho^{2p}}\,.$$

As for the PTR, we find exponential convergence for functions which can be analytically continued to a neighborhood of the interval, with rate depending on the distance to the closest singularity. We note, however, that the PTR is superior for functions which are also periodic [30].

## 3.2 High-order adaptive integration in 1D

Gauss quadrature can be used in a composite manner by dividing an interval of integration into panels, and using a $p$-node rule on each panel. The convergence of such a composite rule is order $2p$, in the sense that if each panel is split in half to form a new composite rule of the same order $p$ with double the number of panels, the error decreases asymptotically by a factor $2^{-2p}$ [31, Sec. 5.2]. This approach allows for the construction of adaptive composite Gauss quadrature rules, in which panels of different lengths are assembled to resolve localized features of a function. An automatic and adaptive algorithm yielding error below a user-specified tolerance $\varepsilon$ is given in Algorithm 2. We note that many variants of 1D adaptive integration have been proposed, and several software packages are available: we refer the reader to [32, Sec. 4.7], [31, Sec. 5.6], and [33,34] for further discussion.

---

**Algorithm 2** Automatic adaptive Gauss quadrature for $f(x)$ on $[a, b]$.

---

**Inputs:** $f(x)$, $a$, $b$, $p$, $\varepsilon$

**Output:** An approximation of $I \equiv \int_a^b f(x)\,dx$

    Apply $p$-node Gauss quadrature to $f$ on $[a, b]$ to obtain $I_0$

    Set $c = (a + b)/2$

    Apply $p$-node Gauss quadrature to $f$ on $[a, c]$ and $[c, b]$ to obtain $I_1$ and $I_2$, respectively

    **if** $|I_0 - (I_1 + I_2)| \leq \varepsilon$ **then**

        Approximate $I$ by $I_1 + I_2$

    **else**

        Call this algorithm on $[a, c]$ and $[c, b]$ with tolerance $\varepsilon$;

    add results to obtain approximation of $I$

    **end if**

**Note:** Since $I_1$ and $I_2$ are used in the recursive call to the algorithm (for example $I_0$, as it appears in the call on $[a, c]$, is equal to $I_1$, as it appears in the original call on $[a, b]$), they can be passed down to avoid repeated work.

---

Algorithm 2 constructs a binary tree of panels on $[a, b]$, and its approximation of $I$ is given by the composite Gauss quadrature rule on the union of the leaf-level panels. The panels automatically refine towards localized features of $f$. As a comparison with the PTR, we consider the example of Fig. 3 with $\eta = 0.01$, for which the PTR of $N = 1000$ points produces an error between $10^{-4}$ and $10^{-3}$. On the other hand, Algorithm 2 with $p = 4$ and $\varepsilon = 10^{-4}$ constructs an adaptive quadrature grid of 256 points which produces an error less than $10^{-6}$. For $\eta = 10^{-4}$ with the same parameters, it constructs a grid of 480 points which produces an error less than $10^{-7}$.

We note that as for Algorithm 1, it is possible for the self-consistency check used in Algorithm 2 to underestimate the true error of $I_0$. However, because of the high-order accuracy of Gauss quadrature, this self-consistency check tends to yield an effective error estimate. In the convergence regime, it is typically a significant overestimate of the error of $I_1 + I_2$, the value ultimately used to approximate the integral $I$. We also note that since the error in the approximation of $I$ is bounded by the sum of the errors made on each leaf-level panel, Algorithm 2 only guarantees $n_{\text{pan}}\varepsilon$ rather than $\varepsilon$ error, where $n_{\text{pan}}$ is the number of leaf-level panels and $\varepsilon$ bounds the error on each panel. This can be straightforwardly remedied by using slightly more sophisticated adaptivity criteria [33], or by adding an additional final step to Algorithm 2: recompute the integral on all leaf-level panels with $p$ increased until convergence below $\varepsilon$.

To resolve an isolated feature which is smooth on the scale $\eta$, Algorithm 2 will refine panels dyadically to yield a smallest panel of width $\mathcal{O}(\eta)$, giving only $\mathcal{O}(\log(\eta^{-1}))$ panels (and hence quadrature nodes) in total.

# 4 Two approaches to adaptive Brillouin zone integration

In this section, we compare two possible generalizations of Algorithm 2 to 2D and 3D, TAI and IAI, and argue that the performance of IAI is superior for integrals of the form (3).

## 4.1 Tree-based adaptive integration

Algorithm 2 can be straightforwardly generalized to integrals on square or cubic domains. Rather than constructing a binary tree of panels and using a Gauss quadrature rule, one builds a quadtree of squares in two dimensions, or an octree of cubes in three dimensions, each using

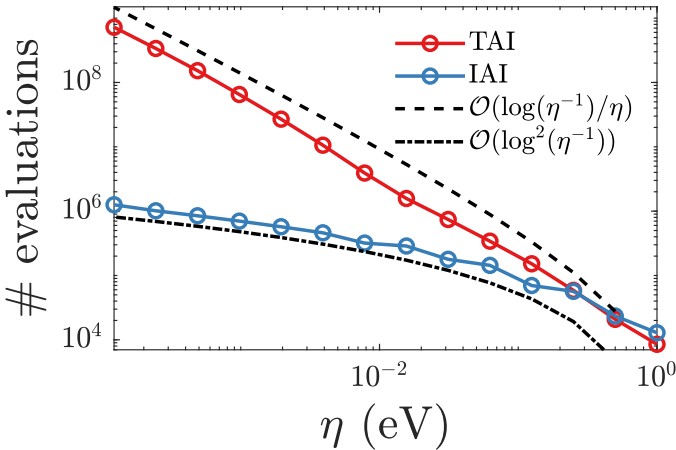

Figure 4: Number of evaluations of the integrand used by TAI and IAI versus $\eta$ for the 2D example calculation described in Fig. 2. Here, we use $\varepsilon = 10^{-5}$, and vary $\eta$.

a Cartesian product of Gauss quadrature rules. Several variants of this basic scheme have been proposed and implemented [35–38].

This approach is incompatible with integration on the IBZ, which is a polygon in 2D and a polytope in 3D. Instead, one can carry out adaptive integration on a tree of triangles or tetrahedra, as in Refs. [22] and [23]. In 3D, this requires partitioning a polytope with a tetrahedral mesh, specifying a splitting strategy which avoids high-aspect ratio tetrahedra, and building a high-order quadrature rule for arbitrary tetrahedra. Refs. [22] and [23] describe meshing and splitting strategies, but use simple, low-order quadrature rules.

A more fundamental issue limits the performance of TAI for integrals of the form (3). Using the term "cell" to refer to squares/cubes/triangles/tetrahedra in a tree-based scheme, we summarize the issue as follows (see the left panel of Fig. 2 for an illustration). The structure of the integrand necessitates resolving features of width $\eta$ along the $(d-1)$-dimensional level set $\det(\omega - H(\boldsymbol{k})) = 0$ on which the integrand of (3) with $\eta = 0^+$ diverges (a curve in 2D and a surface in 3D). Thus, any tree-based adaptive scheme refines to cells of diameter $\mathcal{O}(\eta)$ covering the level set. Since the length or area of the level set is of unit magnitude, and dyadic refinement is also needed in its normal direction, the total number of cells is $\mathcal{O}(\log(\eta^{-1})/\eta^{d-1})$ (in general, for a level set of dimension $\overline{d}$, the number of cells is $\mathcal{O}(\log(\eta^{-1})/\eta^{\overline{d}})$ regardless of $d$).

We verify the $\mathcal{O}(\log(\eta^{-1})/\eta^{d-1})$ scaling of TAI in Fig. 4 for the example shown in Fig. 2. For this calculation, we use the simple TAI scheme described at the beginning of this section. Although it improves on the $\mathcal{O}(\eta^{-d})$ scaling of PTR integration, it is far from optimal, and in practice prevents TAI from accessing the very small $\eta$ regime.

## 4.2 Iterated adaptive integration

A $d$-dimensional integral can be rewritten as a nested sequence of 1D integrals, each of which can be computed using adaptive integration. For example, in 2D,

$$\int_{a_1}^{b_1} \int_{a_2}^{b_2} dx\, dy\, f(x, y) = \int_{a_1}^{b_1} dx\, I_2(x), \quad \text{where} \quad I_2(x) \equiv \int_{a_2}^{b_2} dy\, f(x, y).$$

The integral can therefore be computed by calling Algorithm 2 on the function $I_2$. Within this procedure, $I_2(x)$ can be evaluated for fixed $x$ by calling Algorithm 2 on $f(x, y)$, with $y$ the variable of integration.

Table 1: Asymptotic complexity as $\eta \to 0^+$ of the number of points required to compute $G(\omega)$ in (3), for fixed $\omega$, to a given accuracy, using the three methods described above.

| Method | Complexity |
|--------|-----------|
| PTR | $\mathcal{O}(\eta^{-d})$ |
| TAI | $\mathcal{O}(\log(\eta^{-1})/\eta^{d-1})$ |
| IAI | $\mathcal{O}(\log^d(\eta^{-1}))$ |

For BZ $= [-\pi, \pi]^2$, this IAI method computes (3) as

$$G(\omega) = \int_{-\pi}^{\pi} dk_x \, I_2(k_x), \quad \text{where} \quad I_2(k_x) = \int_{-\pi}^{\pi} dk_y \, \text{Tr}\left[ \left( \omega - H(k_x, k_y) + i\eta \right)^{-1} \right].$$

Fig. 2 illustrates the advantage of IAI. In this case, $H$ is scalar-valued, so the trace vanishes from the integral. The right panel shows the inner integral $I_2(k_x)$. It has only two localized features of width $\mathcal{O}(\eta)$, corresponding to the points at which the curve $H(k_x, k_y) = \omega$ has a tangent aligned with the $k_y$ axis. Thus $I_2(k_x)$ can be obtained using adaptive integration with a grid of $\mathcal{O}(\log(\eta^{-1}))$ points (as discussed at the end of Section 3.2), shown on the horizontal axis. Each evaluation of $I_2(k_x)$ requires computing an integral in $k_y$ with fixed $k_x$. This function also has at most two localized features of width $\mathcal{O}(\eta)$, and therefore can also be adaptively integrated with a grid of $\mathcal{O}(\log(\eta^{-1}))$ points. Thus the cost of computing the integral scales as $\mathcal{O}(\log^2(\eta^{-1}))$. The resulting IAI grid is shown in the middle panel.

This scaling is illustrated in Fig. 4. For $\eta \approx 10^{-4}$, TAI requires over 500 times as many integrand evaluations as IAI. The same argument applies for 3D integrals, leading to $\mathcal{O}(\log^3(\eta^{-1}))$ scaling, so that the advantage of IAI over TAI is even more significant in this case.

We summarize the asymptotic scalings of the three schemes we have discussed in Table 1. Of the two adaptive methods, IAI is almost always preferable, and we therefore do not consider TAI in our numerical tests.

## Rapid evaluation of $H(\mathbf{k})$ in iterated adaptive integration

Since the nearly-singular set varies with $\omega$, adaptive methods must evaluate $H(\mathbf{k})$ on the fly for each fixed $\omega$, a computational bottleneck in certain cases. As discussed in Section 2, we evaluate $H(\mathbf{k})$ using its truncated Fourier series representation (6). This can be done efficiently in IAI by evaluating the series in a sequential manner. We demonstrate this in the 2D case, writing $\mathbf{R} = (m, n)$ and $H_{mn} \equiv H_{\mathbf{R}}$ in (6) to simplify notation:

$$H(k_x, k_y) = \frac{1}{(2\pi)^2} \sum_{m,n=-M/2}^{M/2-1} e^{i(mk_x + nk_y)} H_{mn}.$$

At each step of the outer integration, $k_x$ is fixed, and an inner integration in $k_y$ is carried out. Before the inner integration, we can precompute and store the $M$ numbers

$$H_n(k_x) \equiv \frac{1}{(2\pi)^2} \sum_{m=-M/2}^{M/2-1} e^{imk_x} H_{mn}. \tag{8}$$

Then $H(k_x, k_y)$ can be evaluated at $k_y$ using

$$H(k_x, k_y) = \sum_{n=-M/2}^{M/2-1} e^{ink_y} H_n(k_x). \tag{9}$$

In this way, the cost of evaluating the part of the Fourier series corresponding to the outer integration variable is amortized over all inner quadrature nodes, and becomes negligible. The cost of evaluating $H(\boldsymbol{k})$ in IAI is thus dominated by that of evaluating the 1D Fourier series (9). This can be done efficiently by using the simple recurrence $e^{\pm i n k_y} = e^{\pm i k_y} e^{\pm i(n-1)k_y}$ to compute the exponentials.

### Integration on the irreducible Brillouin zone

Given an explicit Cartesian parameterization of the IBZ, IAI can be used directly on the IBZ by making the straightforward domain replacement

$$\int_{\text{BZ}} dk_x \, dk_y \, dk_z \leftarrow \int_{a_1}^{b_1} dk_x \int_{a_2(k_x)}^{b_2(k_x)} dk_y \int_{a_3(k_x,k_y)}^{b_3(k_x,k_y)} dk_z \,,$$

in 3D, and similarly in 2D. Since the IBZ is a convex polygon/polytope, $a_j$ and $b_j$ are piecewise affine. For example, for the simple cubic BZ $[-\pi, \pi]^3$, the IBZ is a tetrahedron with 1/48th of its volume given by $a_1 = 0$, $b_1 = \pi$, $a_2(k_x) = k_x$, $b_2(k_x) = \pi$, $a_3(k_x, k_y) = 0$, $b_3(k_x, k_y) = k_x$.

In practice, the IBZ may only be described by a collection of point symmetries rather than an explicit parameterization. Ref. [39] describes an algorithm and software package which determines the IBZ as a convex hull characterized by its faces, edges, and vertices. Given this convex hull, it is straightforward to determine $a_1$ and $b_1$, and to evaluate the functions $a_2$, $b_2$, $a_3$, and $b_3$.

We note that the presence of vertices in the IBZ can introduce isolated points of non-differentiability in the integrand, which interferes with high-order convergence. This is straightforwardly remedied by splitting the interval of integration into subintervals determined by these points.

## 5 Frequency domain adaptivity

When $\eta$ is small, $G(\omega)$ itself may develop localized features. A common example is a Van Hove singularity in the $\eta \to 0^+$ limit of $A(\omega)$, defined in (2); when $\eta > 0$, singularities become regularized on a length scale $\mathcal{O}(\eta)$. Automatic adaptive polynomial interpolation is a standard algorithm which can be used to resolve such features, producing an approximation of a function $f(x)$ on an interval $[a, b]$ accurate to a user-specified error tolerance. It operates in a similar manner to Algorithm 2. A polynomial interpolant of degree $p-1$ is built on $[a, b]$ from samples of $f(x)$ at $p$ nodes. The panel is accepted if the estimated error of the interpolant is below a user-specified tolerance, and split otherwise. If the panel is split, polynomial interpolants of degree $p-1$ are similarly constructed on each of the two resulting panels, and the process is repeated. The result is a piecewise degree $p-1$ polynomial interpolant on a collection of panels which are automatically adapted to the localized features of $f(x)$. The interpolant can be evaluated rapidly at any given point $x$ by descending the binary tree produced by the procedure to identify the leaf-level panel containing $x$, and then evaluating the polynomial interpolant corresponding to that panel. The interpolant is uniformly accurate to the specified tolerance.

A complete description of the procedure requires identifying specific polynomial interpolation and error estimation methods. For the former, we use barycentric interpolation at Chebyshev nodes, which is fast, stable, well-conditioned, and spectrally accurate [40, 41]. There are several possible error estimation strategies. A fairly robust approach is to compare an interpolant on $[a, b]$ to interpolants on $[a, (a + b)/2]$ and $[(a + b)/2, b]$ on a dense grid of points, and estimate the error as the maximum absolute discrepancy. A slightly less robust but

commonly used approach is to use the sum of the absolute values of the last few coefficients of the Chebyshev expansion on a panel as an error estimate. This method is often effective in practice, and requires fewer evaluations of $f(x)$, so we use it for the calculations carried out in the next section.

This automatic adaptive Chebyshev interpolation procedure can be applied to obtain a piecewise polynomial representation of the spectral function $A(\omega)$ on a given interval $[\omega_{\min}, \omega_{\max}]$. The interpolants on each panel are constructed from samples of $A(\omega)$ at particular values of $\omega$, and these samples can be obtained using either the automatic PTR method described in Section 2.2, or IAI. To resolve localized features at a scale $\mathcal{O}(\eta)$, the adaptive interpolant requires $\mathcal{O}(\log(\eta^{-1}))$ evaluations of $A(\omega)$, each of which requires computing a BZ integral. The expected scaling of the spectral function calculation as $\eta \to 0^+$ is therefore $\mathcal{O}(\log(\eta^{-1})/\eta^3)$ using the automatic PTR method and $\mathcal{O}(\log^4(\eta^{-1}))$ using IAI.

# 6 Example: Spectral function of strontium vanadate

We demonstrate the automatic IAI and PTR methods, as well as the frequency adaptivity procedure, by calculating the spectral function for the correlated metal strontium vanadate, $SrVO_3$. Its cubic structure (with space group $Pm\bar{3}m$ (221)) and the isolated set of three $t_{2g}$-derived low-energy states results in a simple tight-binding-like electronic structure of three degenerate orbitals, yielding $3 \times 3$ matrices $H(\boldsymbol{k})$. $SrVO_3$ is known to exhibit Fermi liquid behavior [42], with the typical scattering rate scaling given in (4). For now, we consider a scattering rate $\eta$ which is constant in frequency, but expect our algorithms to be particularly useful for frequency-dependent self-energies.

We construct the tight-binding Hamiltonian $H_{\boldsymbol{R}}$ in (6) from the low-lying $t_{2g}$ manifold by projecting the ab initio Hamiltonian onto the three partially-filled vanadium $3d$ orbitals using the WANNIER90 code [7]. Using the planewave-based QUANTUM ESPRESSO package [43], we evaluate the ab initio Hamiltonian at $M = 10$ equispaced grid points per dimension to obtain the samples $H(q_m)$ referred to in Section 2.1. We use default parameters for the ground state calculation, with lattice parameter $a = 3.859$ Å, and the Perdew–Burke–Ernzerhof functional [44] with standard scalar-relativistic ultrasoft pseudopotentials [45].

We compute the spectral function as defined in (2) and (3) with $d = 3$. We carry out the integration over the IBZ, which contains 48 point symmetries, using the methods described in Appendix B and Section 4.2 for the PTR and IAI, respectively. We set $\varepsilon = 10^{-5}$ for both the automatic PTR method and IAI, and use $p = 4$ Gauss quadrature nodes per panel for IAI to obtain eighth-order accurate quadratures. All calculations are performed on a single core of a workstation with an Intel Xeon Gold 6244 processor and 256 GB of RAM.

In Fig. 5, we show wall clock timings for the calculation of the spectral function at the Fermi level $\omega = 0$ eV, for several values of $\eta$ down to the sub-meV scale. We observe the expected asymptotic scalings. The automatic PTR method could not be used for $\eta \lesssim 4$ meV due to memory constraints, specifically that of storing $H(\boldsymbol{k})$ on a fine equispaced grid. $H(\boldsymbol{k})$ could be evaluated on the fly as in IAI, rather than computing and storing it once, but when $A(\omega)$ must be computed at more than one value of $\omega$, this significantly increases the computational cost. By contrast, IAI has modest memory requirements which do not increase significantly as $\eta$ is decreased.

We next compute the full spectral function $A(\omega)$ using the adaptive interpolation method described in Section 5. We use a tolerance $10^{-4} = 10\varepsilon$, and a Chebyshev interpolant of $q = 16$ nodes on each panel. The result is shown in Fig. 6 for several values of $\eta$, along with the final adaptive frequency grid obtained by the automatic algorithm for $\eta \approx 1$ meV. We observe multiple Van Hove band edges, as well as a stronger feature at $\omega \approx 0.9$ eV corresponding to flat

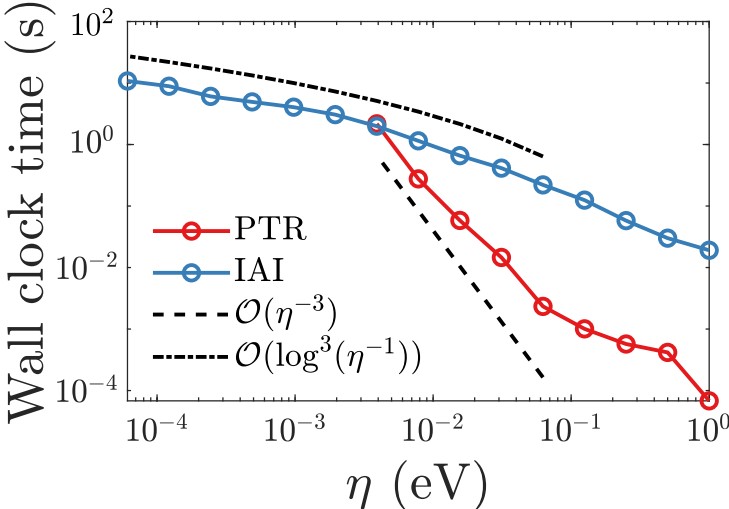

Figure 5: Wall clock time versus $\eta$ for the calculation of the spectral function $A(\omega)$ of SrVO$_3$ at $\omega = 0$ eV, comparing the automatic PTR and IAI methods. For the automatic PTR, we only include the costs of summation for the final values of $N = N_{\text{trial}}$ and $N = N_{\text{test}}$ chosen by the algorithm, since other costs, particularly that of evaluating $H(\boldsymbol{k})$ on the PTR grid, would be amortized over many frequencies in a calculation of the full spectral function.

bands. The adaptive interpolation algorithm refines into all localized features, as expected. The maximum error, measured against a converged reference for each $\eta$ obtained by repeating all calculations with $\varepsilon = 10^{-8}$, is at most $7.4 \times 10^{-5}$ using the automatic PTR method and $4.7 \times 10^{-4}$ using the IAI method.

We plot wall clock timings for both methods in Fig. 7, and observe the expected scaling with $\eta$. IAI is faster than the automatic PTR method for $\eta \lesssim 15$ meV, and an optimal scheme would switch from one method to the other at this point. Table 2 shows the number of PTR grid points per dimension used by the automatic PTR algorithm to obtain the specified accuracy for each $\eta$. The automatic PTR method cannot be used for $\eta \lesssim 4$ meV due to memory constraints, and at $\eta \approx 4$ meV, the finest required grid uses $N = 2714$ points per dimension. For $\eta \approx 1$ meV, computing $A(\omega)$ using IAI takes just under 25 minutes. Extrapolating from the data shown,

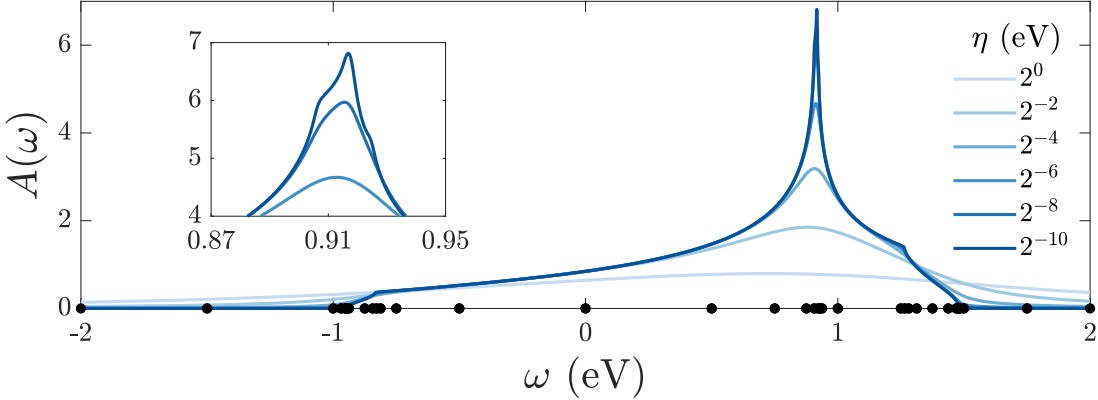

Figure 6: Spectral function $A(\omega)$ of SrVO$_3$ with values of $\eta$ varying from 1 eV to $2^{-10}$ eV $\approx 1$ meV. The inset shows details of the region near an approximate singularity, regularized on the scale $\eta$.

and ignoring memory limitations, the automatic PTR method would have taken over 4.5 days to complete the same calculation—over 250 times as long as IAI—and would have required over $N = 10\,000$ points per dimension. Although the IAI calculation used a single core, several straightforward options for parallelization are available. For example, a simple parallelization over frequency points would yield this result in a few minutes on a modern workstation.

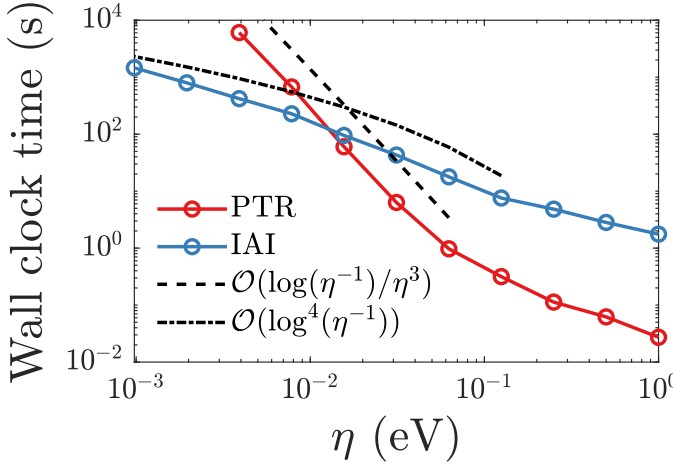

Figure 7: Wall clock time versus $\eta$ for the calculation of the spectral function $A(\omega)$ of SrVO$_3$, using adaptive frequency interpolation with both the automatic PTR method and IAI.

Table 2: Number $N$ of PTR grid points per dimension used for the integration error tolerance $\varepsilon = 10^{-5}$ to compute the spectral function of SrVO$_3$ shown in Fig. 6.

| $\eta$ (eV) | $2^0$ | $2^{-1}$ | $2^{-2}$ | $2^{-3}$ | $2^{-4}$ | $2^{-5}$ | $2^{-6}$ | $2^{-7}$ | $2^{-8}$ |
|---|---|---|---|---|---|---|---|---|---|
| $N$ | 65 | 112 | 124 | 148 | 196 | 340 | 678 | 1356 | 2714 |

## 7 Conclusion

We have presented automatic high-order accurate methods for BZ integration with positive broadening, and adaptive frequency sampling. Our algorithms are straightforward to implement and deliver results accurate to a user-specified error tolerance $\varepsilon$. For integration, we recommend the automatic PTR method when $\eta$ is not too small, and IAI otherwise. Several possible heuristics could be used to choose between the two methods based on $\eta$ and $\varepsilon$, and we will investigate these in our future work. For adaptive calculations, we have shown that IAI is preferable to TAI, because of its superior asymptotic scaling as $\eta \to 0^+$ and algorithmic efficiency, as well as its simplicty of implementation for IBZ calculations. In particular, for integrals of the form (3), IAI has the mild asymptotic complexity $\mathcal{O}(\log^d(1/\eta))$.

Although for concreteness we have restricted our discussion to BZ integrals for Green's functions of the form (3), and have demonstrated the performance of our algorithms in calculations of the spectral function with constant, diagonal self-energy $\Sigma = -i\eta$, our framework is fairly general. In particular, the case of Green's functions involving frequency-dependent and non-diagonal self-energies is straightforward, and other possible applications include BZ integrals arising in response functions. The generalization to momentum-dependent self-energies $\Sigma(\boldsymbol{k}, \omega)$ requires that $\Sigma$ can be evaluated rapidly at arbitrary points $(\boldsymbol{k}, \omega)$. Although the

given representation of $\Sigma$ may be expensive to evaluate, it can be replaced by an approximant which can be evaluated quickly (e.g., a piecewise polynomial interpolant) in a precomputation. The proper approximant depends on the underlying structure of $\Sigma$, and is therefore problem-dependent.

## Acknowledgments

We thank J. Bonini, S. Poncé, J.-M. Lihm, M. Ghim, C.-H. Park, A. Togo, A. Hampel, C. Dreyer, A. Levitt, and E. Letournel for helpful discussions. We also thank S. Tsirkin for pointing out the algorithm described in Appendix D, and for several other useful suggestions on the manuscript. The Flatiron Institute is a division of the Simons Foundation.

## A    Convergence of the periodic trapezoidal rule for Brillouin zone integrals

In this Appendix, we develop convergence results for the PTR which extend the discussion in Section 2. Given analytic $H(\boldsymbol{k})$, we would like a practical lower bound on $a$, the convergence rate appearing in an error estimate $e_N \leq Ce^{-aN}$ for the $d$-dimensional PTR with $N$ points per dimension (as in Theorem 1 for 1D). Such bounds can be used to guide the selection of $\Delta N$ in Algorithm 1 in order to make the error estimate as robust as possible. Algorithm 1 estimates the error of the $N^d$-point PTR as $|e_N - e_{N+\Delta N}|$. If $e_{N+\Delta N}$ is smaller than $e_N$ by a factor $\gamma \gg 1$, then this error estimate is accurate with high probability. Thus we must choose $\Delta N$ such that

$$\gamma \leq e_N/e_{N+\Delta N} \approx e^{a\Delta N} \implies \Delta N \geq \log(\gamma)/a. \tag{A.1}$$

To justify such an exponential error estimate, we first generalize Theorem 1 to $d > 1$. We then give a rigorous lower bound on its rate $a$, in terms of the width of the strip of analyticity, in each dimension, of the integrand of (3). The bound takes the form $a \geq C\eta$, where $C$ depends on an upper bound of $\nabla_{\boldsymbol{k}} H(\boldsymbol{k})$, in a norm to be specified for the case of scalar-valued $H(\boldsymbol{k})$, and conjectured otherwise. If this bound is on the order of 1, then (A.1) with $\gamma = 10$ gives $\Delta N \gtrsim 2.3/\eta$. We have checked that this is the case for the example considered in Section 6, and therefore use $\Delta N = 2.3/\eta$ for our numerical experiments.

**Generalization of Theorem 1 to dimension $d$**

For simplicity, we state the result for 2D, but the same technique gives an analogous result in 3D.

**Theorem 3.** *Let $f(x, y)$ be doubly $2\pi$-periodic. Suppose that for each fixed $y = y_0$, $f(x, y_0)$ is analytic in the strip $|\operatorname{Im} x| < a_1$, with $|f(x, y_0)| \leq C_1$, and for each fixed $x = x_0$, $f(x_0, y)$ is analytic in the strip $|\operatorname{Im} y| < a_2$, with $|f(x_0, y)| \leq C_2$. Then for any $N_1, N_2 \geq 1$,*

$$\left| \int_{[0,2\pi]^2} f(x, y) \, dx \, dy - \frac{(2\pi)^2}{N_1 N_2} \sum_{n_1, n_2 = 0}^{N_1 - 1, N_2 - 1} f\left( \frac{2\pi n_1}{N_1}, \frac{2\pi n_2}{N_2} \right) \right| \leq 8\pi^2 \left( \frac{C_1}{e^{a_1 N_1} - 1} + \frac{C_2}{e^{a_2 N_2} - 1} \right).$$

*Proof.* The triangle inequality gives

$$\left| \int_{[0,2\pi]^2} dx\, dy\, f(x,y) - \frac{(2\pi)^2}{N_1 N_2} \sum_{n_1,n_2=0}^{N_1-1,N_2-1} f\left(\frac{2\pi n_1}{N_1}, \frac{2\pi n_2}{N_2}\right) \right|$$

$$\leq \left| \int_0^{2\pi} dx \left[ \int_0^{2\pi} dy\, f(x,y) - \frac{2\pi}{N_2} \sum_{n_2=0}^{N_2-1} f\left(x, \frac{2\pi n_2}{N_2}\right) \right] \right|$$

$$+ \left| \frac{2\pi}{N_2} \sum_{n_2=0}^{N_2-1} \int_0^{2\pi} dx\, f\left(x, \frac{2\pi n_2}{N_2}\right) - \frac{(2\pi)^2}{N_1 N_2} \sum_{n_1,n_2=0}^{N_1-1,N_2-1} f\left(\frac{2\pi n_1}{N_1}, \frac{2\pi n_2}{N_2}\right) \right| \equiv T_1 + T_2.$$

Fixing $x$ in the inner integral of $T_1$ and applying Theorem 1 gives

$$T_1 \leq \int_0^{2\pi} dx\, \frac{4\pi C_2}{e^{a_2 N_2} - 1} = \frac{8\pi^2 C_2}{e^{a_2 N_2} - 1},$$

for any $N_2 \geq 1$. We then have

$$T_2 = \left| \frac{2\pi}{N_2} \sum_{n_2=0}^{N_2-1} \left[ \int_0^{2\pi} dx\, f\left(x, \frac{2\pi n_2}{N_2}\right) - \frac{2\pi}{N_1} \sum_{n_1=0}^{N_1-1} f\left(\frac{2\pi n_1}{N_1}, \frac{2\pi n_2}{N_2}\right) \right] \right|$$

$$\leq \frac{2\pi}{N_2} \sum_{n_2=0}^{N_2-1} \frac{4\pi C_1}{e^{a_1 N_1} - 1} = \frac{8\pi^2 C_1}{e^{a_1 N_1} - 1},$$

for any $N_1 \geq 1$, where we have fixed $y = \frac{2\pi n_2}{N_2}$ in the inner integral and applied Theorem 1. $\quad\square$

Setting $N_1 = N_2$ and taking $a = \min(a_1, a_2)$, $C = \max(C_1, C_2)$, we obtain the error bound $\frac{16\pi^2 C}{e^{aN} - 1}$, which is of the form required to justify the argument underlying (A.1). In $d$ dimensions, the bound is $\frac{2d(2\pi)^d C}{e^{aN} - 1}$.

**Lower bound on the analytic strip width $a$**

We begin with the simplest case of (3): $d = 1$ with $H(k)$ scalar-valued. The following argument gives some intuition for the discussion. Let $k_0 \in \mathbb{R}$ be such that $H(k_0) = \omega$. If $\eta$ is small, we expect a pole of the integrand near $k_0$ in the complex plane. A Taylor expansion about $k = k_0$ approximates the pole location as

$$k = k_0 + \frac{i\eta}{H'(k_0)}.$$

We therefore expect that $(\omega - H(k) + i\eta)^{-1}$ is analytic in a strip $|\operatorname{Im} k| < a$ of width

$$a \gtrsim \frac{\eta}{\max_k |H'(k)|}. \tag{A.2}$$

In 2D, in order to estimate a lower bound on the analytic strip widths mentioned in Theorem 3, we can repeat the argument with $k \leftarrow k_x$ for each fixed $k_y$, and similarly with $k \leftarrow k_y$ for each fixed $k_x$. We obtain

$$a_1 \gtrsim \frac{\eta}{\max_{\boldsymbol{k}} \left| \frac{\partial H}{\partial k_x}(k_x, k_y) \right|}, \qquad a_2 \gtrsim \frac{\eta}{\max_{\boldsymbol{k}} \left| \frac{\partial H}{\partial k_y}(k_x, k_y) \right|}.$$

We can follow a similar procedure in 3D.

The following result clarifies this intuition more rigorously, starting with 1D. We absorb $\omega$ into $H$ as it does not affect the result.

**Lemma 1.** *Let $H$ be a $2\pi$-periodic real analytic function on $\mathbb{R}$, and let $\eta > 0$. Suppose $a > 0$ is such that i) $H$ can be analytically continued throughout the closed strip $|\mathrm{Im}\, z| \le a$, and ii) $\max_{|\mathrm{Im}\, z| \le a} |H'(z)| < \eta/a$. Then $(H(z) + i\eta)^{-1}$ is analytic and bounded in this closed strip.*

*Proof.* Since $H(z)$ is analytic in the strip $|\mathrm{Im}\, z| \le a$, $(H(z) + i\eta)^{-1}$ is also analytic there except possibly at isolated poles. Note that the case of constant $H \equiv -i\eta$ is not possible since $H$ is real on $\mathbb{R}$. Suppose $z_p$ is such a pole, i.e., $H(z_p) = -i\eta$ with $\left|\mathrm{Im}\, z_p\right| \le a$. Then

$$-i\eta - H(x_p) = \int_{x_p}^{z_p} H'(z)\, dz\,,$$

with $x_p = \mathrm{Re}\, z_p$. Bounding the magnitudes of both sides gives $\eta \le |i\eta + H(x_p)| \le a \max_{|\mathrm{Im}\, z| \le a} |H'(z)|$, since $H(x_p) \in \mathbb{R}$. This contradicts the hypothesis ii). Therefore $(H(z) + i\eta)^{-1}$ is analytic and bounded in the closed strip.

$\square$

This result can be used to obtain a practical lower bound for the rate of exponential decay provided by Theorem 1 in the present setting, in which $H(k)$ is a truncated Fourier series, and therefore entire. We define $M(\alpha) \equiv \max_{|\mathrm{Im}\, z| \le \alpha} \left|H'(z)\right|$, a continuous, non-decreasing function of $\alpha \ge 0$ which may be readily computed. Using the bisection algorithm, we can then find a height $\alpha^* > 0$ as large as possible such that $\alpha^* M(\alpha^*) < \eta$. The hypotheses of Lemma 1 are satisfied with $a$ replaced by $\alpha^*$, so we find that the exponential rate of decay $a$ appearing in Theorem 1 satisfies

$$a \ge \frac{\eta}{M(\alpha^*)} = \frac{\eta}{\max_{|\mathrm{Im}\, z| \le \alpha^*} |H'(z)|}\,.$$

Since $M(\alpha^*) \to \max_k \left|H'(k)\right|$ as $\eta \to 0^+$, we see that (A.2) is a reasonable approximation in this limit.

In 2D, we can define $M_1(\alpha_1) \equiv \max_{k_y} \max_{|\mathrm{Im}\, z| \le \alpha_1} \left|\frac{\partial H}{\partial k_x}(z, k_y)\right|$, and $\alpha_1^* > 0$ a height as large as possible with $\alpha_1^* M_1(\alpha_1^*) < \eta$. $M_2(\alpha_2)$ and $\alpha_2^*$ can be defined similarly by switching the roles of $k_x$ and $k_y$. We then obtain the lower bounds

$$a_1 \ge \frac{\eta}{M_1(\alpha_1^*)} = \frac{\eta}{\max_{k_y} \max_{|\mathrm{Im}\, z| \le \alpha_1^*} \left|\frac{\partial H}{\partial k_x}(z, k_y)\right|}\,,$$

$$a_2 \ge \frac{\eta}{M_2(\alpha_2^*)} = \frac{\eta}{\max_{k_x} \max_{|\mathrm{Im}\, z| \le \alpha_2^*} \left|\frac{\partial H}{\partial k_y}(k_x, z)\right|}\,,$$

which can then be inserted into Theorem 3.

Rigorous convergence rate bounds for the case of matrix-valued $H(\boldsymbol{k})$ are more challenging, and we leave them for future work. For $d = 1$, we conjecture that the arguments above will be valid with $\left|H'(z)\right|$ replaced by $\left\|H'(z)\right\|_2$. The $d > 1$ case might then be treated in a similar manner as above.

## B  Symmetrized periodic trapezoidal rule in the irreducible Brillouin zone

Let $S_1, \ldots, S_p$ be the orthogonal matrices representing the point symmetries of the lattice, so that $H(S_j \boldsymbol{k}) = H(\boldsymbol{k})$. We define $W$, an irreducible wedge of the BZ, as the closed set such that

$BZ = \cup_{j=1}^{p} S_j W$ and $\cap_{j=1}^{p} \left(S_j W\right)^{\mathrm{o}} = \varnothing$. Denoting the integrand of (3) by $f\left(H(\boldsymbol{k})\right)$, and making the change of variables $\boldsymbol{k} \leftarrow S_i \boldsymbol{k}$, we can calculate (3) as an integral over $W$:

$$\int_{BZ} d^d\boldsymbol{k}\, f\left(H(\boldsymbol{k})\right) = \sum_{j=1}^{p} \int_{S_j W} d^d\boldsymbol{k}\, f\left(H(\boldsymbol{k})\right) = \sum_{j=1}^{p} \int_W d^d\boldsymbol{k}\, f\left(H(S_j\boldsymbol{k})\right) = p \int_W d^d\boldsymbol{k}\, f\left(H(\boldsymbol{k})\right). \quad \text{(B.1)}$$

The PTR cannot be applied directly to the integral over $W$, since it is not a regular domain on which $H(\boldsymbol{k})$ is periodic. However, under some constraints, the $d$-dimensional PTR can be modified to take advantage of the point symmetries. In particular, if we take the number of grid points in each dimension to be equal (e.g. $N_1 = N_2$ in (7)), then the PTR sum can be converted exactly into a sum over only the grid points of the PTR lying in the IBZ, suitably reweighted.

Indeed, in this case, the PTR grid respects the point symmetries. Let $S$ be a particular point symmetry. Since $S$ maps any reciprocal lattice vector to another reciprocal lattice vector, we have $SB = BM$ for some $M \in \mathbb{Z}^{d \times d}$. Consider a point $\boldsymbol{g} = B\boldsymbol{n}/N$ of the $N^d$-point PTR lying in $W$, $\boldsymbol{n} \in \mathbb{Z}^d$ with $0 \leq n_1, \ldots, n_d \leq N-1$. We have $S\boldsymbol{g} = BM\boldsymbol{n}/N \in SW$, an integer linear combination of the vectors $B/N$, i.e., a PTR grid point in $SW$. Therefore all PTR grid points in $W$ map to PTR grid points in $SW$, and all PTR grid points in $SW$ are the image under $S$ of a grid point in $W$, as can be seen by repeating the argument on $SW$ with the symmetry $S^{-1}$. As a consequence, one can reduce the full PTR sum to a sum over the grid points $\boldsymbol{g} \in W$, weighted by $w_{\boldsymbol{g}} = \left| \cup_{i=1}^{p} S_i \boldsymbol{g} \right|$, the number of distinct images of $\boldsymbol{g}$ in the BZ under the point symmetries. This number is equal to $p$ for interior points of the IBZ, and less than $p$ for boundary points.

Letting $K = \{B\boldsymbol{n} \mid 0 \leq n_1, \ldots, n_d \leq N-1, B\boldsymbol{n} \in W\}$ be the collection of PTR grid points in $W$ (or, more generally, a minimal sufficient set of grid points which may not all fall in a single irreducible wedge), we obtain an analogue of (B.1) for the $N^d$-point PTR:

$$\frac{|\det B|}{N^d} \sum_{n_1,\ldots,n_d=0}^{N-1} f\left(H(B\boldsymbol{n})\right) = \frac{|\det B|}{N^d} \sum_{\boldsymbol{g} \in K} w_{\boldsymbol{g}} f\left(H(\boldsymbol{g})\right).$$

The set $K$ and weights $w_{\boldsymbol{g}}$ can be computed by the following algorithm, which does not require a parameterization of the IBZ, but only the point symmetries.

---

**Algorithm 3** Determine grid points, weights of symmetrized $N^d$-point PTR.

---

**Inputs:** $N, S_1, \ldots, S_p$
**Outputs:** $K, w_{\boldsymbol{g}}$

Initialize $K$ as the full $N^d$-point PTR grid
**loop** over remaining points $\boldsymbol{g} \in K$
    $w_{\boldsymbol{g}} = 1$
    **for** $j = 2, \ldots, p$ **do**
        **if** $S_j \boldsymbol{g} \in K \wedge S_j \boldsymbol{g} \neq \boldsymbol{g}$ **then**
            $K \leftarrow K \setminus \{S_j \boldsymbol{g}\}$
            $w_{\boldsymbol{g}} \leftarrow w_{\boldsymbol{g}} + 1$
        **end if**
    **end for**
**end loop**
**Note:** We assume here that $S_1$ is the identity map.

---

In some cases, the PTR can be symmetrized under weaker constraints than that considered here. For example, the 2D PTR respects all symmetries of any non-square rectangular lattice even if a different number of grid points is chosen in each dimension. It may also be possible

to take advantage of partial symmetries. For example, the PTR respects half of the symmetries of the square lattice if a different number of grid points is used in each dimension, and can therefore be symmetrized to a domain consisting of two irreducible wedges. We do not attempt to classify all such possibilities.

## C Efficient evaluation of $H(\boldsymbol{k})$ for periodic trapezoidal rule

In our PTR integration procedure, we compute $H(\boldsymbol{k})$ at all grid points before summing. This could be done using a zero-padded $N^d$-point FFT, but that is typically not the most efficient approach for the present case in which $N \gg M$. Instead, we evaluate the Fourier series directly using the iterated summation approach summarized in (8) and (9): we loop over grid points dimension-by-dimension, precomputing Fourier series coefficients for fixed components of $\boldsymbol{k}$ so that that dominant cost is evaluating a truncated 1D Fourier series of $M$ terms at each point. For a full BZ calculation, the total cost scales as $\mathcal{O}(MN^d)$. For the symmetrized PTR on the IBZ, described in Appendix B, some care must be taken to implement iterated summation only over the points in $K$. In particular, for each index $n_i$ in an outer sum, one must keep track of the the range of indices $n_{i+1}, \dots, n_d$ in each inner sum corresponding to points included in $K$. The cost of the resulting algorithm scales as $\mathcal{O}(M |K|) = \mathcal{O}(MN^d/p)$.

An alternative approach is the pruned FFT algorithm [46], described in the context of BZ integration in Ref. [47]. The pruned FFT, which has $\mathcal{O}(N^d \log M)$ complexity, reconstructs the $N^d$-point Fourier transform from a collection of rephased $M^d$-point FFTs. When $M$ is small, it is not clear whether or not this reduced complexity translates to a performance improvement over direct evaluation of the Fourier series. We therefore defer a comparison with the pruned FFT to future work.

## D Alternative automatic PTR algorithm for fixed frequency grids

If a collection $\omega_j$ of frequencies at which to evaluate $G(\omega)$ is specified in advance, there is an alternative to the automatic PTR algorithm described in Section 2.2 which has two advantages: (1) it avoids storing $H(\boldsymbol{k})$ on a PTR grid, and (2) it avoids the possibility of computing $G(\omega_j)$ on a denser PTR grid than is necessary for a given frequency $\omega_j$. This procedure is described in Algorithm 4. We denote the integrand of (3) by $f(H(\boldsymbol{k}), \omega)$, and points in the PTR grid by $\boldsymbol{g}$. The algorithm is simultaneously described for the unsymmetrized PTR for full BZ calculations, and the symmetrized PTR described in Appendix B, with the option of using a symmetrized grid denoted by writing "symmetrized" in parentheses. We denote the weights in either PTR grid by $w_{\boldsymbol{g}}$. For the unsymmetrized $N^d$-point grid on the full BZ $[-\pi, \pi]^d$, we have $w_{\boldsymbol{g}} = (2\pi/N)^d$, and for the symmetrized PTR grid, the weights can be computed using Algorithm 3.

Algorithm 4 cannot be used if the evaluation frequencies $\omega_j$ are not specified in advance, as is the case when the adaptive frequency interpolation scheme described in Section 5 is used. Since use of an adaptive frequency grid gives a reduction in the number of points required to resolve $G(\omega)$ from $\mathcal{O}(\eta^{-1})$ to $\mathcal{O}(\log(\eta^{-1}))$ compared with a uniform grid, the advantages provided by Algorithm 4 are typically insufficient to justify its use if it is necessary to fully resolve $G(\omega)$, rather than evaluate it on a pre-specified grid.

---

**Algorithm 4** Automatic PTR to compute $G(\omega_j)$ at a collection of frequencies $\omega_j$.

---

**Inputs:** $\eta > 0$, $\varepsilon > 0$, positive integers $N$ and $\Delta N$, $n_\omega$ points $\omega_j$, $H_R$
**Output:** $G(\omega_j)$, $j = 1, \ldots, n_\omega$, correct to $\varepsilon$ accuracy

    Initialize $J = \{1, \ldots, n_\omega\}$
    Initialize $G_{\text{trial}}(\omega_j) = 0$ for all $j \in J$
    Compute $w_g$ for (symmetrized) $N^d$-point PTR grid
    **loop** over $g$ in (symmetrized) $N^d$-point PTR grid
        Evaluate $H(g)$
        **for** $j = 1, \ldots, n_\omega$ **do**
            $G_{\text{trial}}(\omega_j) \mathrel{+}= w_g f(H(g), \omega_j)$
        **end for**
    **end loop**
    **while** $J \neq \emptyset$ **do**
        $N \leftarrow N + \Delta N$
        Compute $w_g$ for (symmetrized) $N^d$-point PTR grid
        Initialize $G_{\text{test}}(\omega_j) = 0$ for all $j \in J$
        **loop** over $g$ in (symmetrized) $N^d$-point PTR grid
            Evaluate $H(g)$
            **for** $j \in J$ **do**
                $G_{\text{test}}(\omega_j) \mathrel{+}= w_g f(H(g), \omega_j)$
            **end for**
        **end loop**
        **for** $j \in J$ **do**
            $e \leftarrow \left| G_{\text{trial}}(\omega_j) - G_{\text{test}}(\omega_j) \right|$
            **if** $e < \varepsilon$ **then**
                Set $G(\omega_j) = G_{\text{test}}(\omega_j)$
                $J \leftarrow J \setminus \{j\}$
            **else**
                $G_{\text{trial}}(\omega_j) \leftarrow G_{\text{test}}(\omega_j)$
            **end if**
        **end for**
    **end while**

---

# E   Linear tetrahedron method with $\eta > 0$

The LTM is a popular BZ integration method in electronic structure calculations, and we therefore discuss it and compare it with our approach in this Appendix. We first emphasize that the classical use case of the LTM is for BZ integrals with integrands involving large Hamiltonian matrices which cannot be efficiently evaluated on the fly using Wannier interpolation, and with zero broadening $\eta = 0^+$. This is a challenging setting, in which the LTM has significant strengths, but is not the focus of this article. In the setting of a Wannier-interpolated Hamiltonian $H(k)$ that can be evaluated efficiently on the fly, and a non-zero but possibly small broadening $\eta > 0$ (or, more broadly, a self-energy), more research is necessary to understand how the LTM should be implemented, whether it is robust, and what its asymptotic cost and performance are as $\eta \to 0^+$.

    The idea of the LTM for integrals of the form (3) can be summarized as follows: the BZ is partitioned into tetrahedra of diameter approximately $\Delta k$, $H(k)$ is replaced on each tetrahedron by a linear interpolant through its values at the vertices, and the resulting integrals are computed analytically. If $H(k)$ is scalar-valued, then the required analytical formula can be

derived, even for $\eta = 0^+$, and if it is matrix-valued, then the scalar formula can be applied to its diagonalization.

The LTM has the low-order accuracy $\mathcal{O}((\Delta k)^2)$, a significant limitation to its use as a black-box method to efficiently deliver a user-specified accuracy. More importantly, in the case of (1) with a non-zero self-energy, diagonalizing $H(\boldsymbol{k}) + \Sigma(\omega)$ yields complex eigenvalues, and it is unclear how the LTM should be implemented in that case. Ref. [15] describes the issue in terms of an ambiguity in the ordering of complex eigenvalues in the vicinity of band crossings, and proposes a heuristic algorithm to address the problem, but the convergence properties of the resulting scheme are not known.

Even if we focus on the case of (3) with scalar-valued $H(\boldsymbol{k})$, questions of robustness and efficiency remain. To illustrate one such concern, we show that in the classical case $\eta = 0^+$, the LTM can yield a density of states with arbitrarily large error at some $\omega$, no matter how small $\Delta k$ is, i.e., the LTM does not converge uniformly. We will then return to the present case of interest, $\eta > 0$.

Consider the calculation of the density of states

$$A_0(\omega) = \int_{\text{BZ}} d^3\boldsymbol{k}\, \delta\left(\omega - H(\boldsymbol{k})\right),$$

the $\eta = 0^+$ limit of (2), for the simple case $H(\boldsymbol{k}) = \cos(k_x) + \cos(k_y) + \cos(k_z)$. $A_0(\omega)$ has a Van Hove singularity at $\omega_0 = 3$ due to the extremum at $\boldsymbol{k} = \boldsymbol{0}$. It is straightforward to show that $A_0(\omega) \sim \sqrt{\omega_0 - \omega}$ as $\omega \to \omega_0^-$. For a fixed partition of BZ by tetrahedra of diameter $\Delta k$, consider $\omega$ sufficiently close to $\omega_0$ that

$$A_0(\omega) = \int_T d^3\boldsymbol{k}\, \delta\left(\omega - H(\boldsymbol{k})\right),$$

for a single tetrahedron $T$, i.e., no other tetrahedron contributes to $A_0(\omega)$. Within $T$, the LTM makes the approximation $H(\boldsymbol{k}) \approx \boldsymbol{v} \cdot \boldsymbol{k} + \alpha$. This yields

$$A_0(\omega) \approx \int_T d^3\boldsymbol{k}\, \delta\left(\omega - (\boldsymbol{v} \cdot \boldsymbol{k} + \alpha)\right) = \frac{s(\omega)}{|\boldsymbol{v}|},$$

with

$$s(\omega) = \int_{\{\boldsymbol{k} \in T \,|\, \boldsymbol{v} \cdot \boldsymbol{k} + \alpha = \omega\}} dS(\boldsymbol{k}),$$

the surface area of the plane $\boldsymbol{v} \cdot \boldsymbol{k} + \alpha = \omega$ in $T$. If $\omega$ is chosen so that $s(\omega) > 0$, then the resulting approximation of $A_0(\omega)$, which should be $\sim \sqrt{\omega_0 - \omega}$, can be arbitrarily large, depending on how small $|\boldsymbol{v}|$ is. $|\boldsymbol{v}|$ is a measure of the flatness of the linear interpolant of $H(\boldsymbol{k})$ through the vertices of $T$, and therefore depends sensitively on the specific location of $T$ relative to $\boldsymbol{k} = \boldsymbol{0}$. By simple translation of the mesh, one can arrange $|\boldsymbol{v}|$ to be arbitrarily small, leading to an arbitrarily large spurious peak in $A_0(\omega)$. This phenomenon may be encountered in the vicinity of any band extremum, and the tetrahedral mesh cannot in general be arranged to avoid it without significant customization of the mesh for each given band structure. Although the range in $\omega$ over which such a problem is encountered decreases as $\Delta k$ decreases, the expected magnitude of the error increases, since $H(\boldsymbol{k})$ becomes flatter in the tetrahedron containing the band edge. The problem persists no matter how small $\Delta k$ is taken.

For the $\eta = 0^+$ case, the LTM cannot therefore be used to build a robust and automatic BZ integration algorithm providing results to a user-specified accuracy without correcting failure modes of this nature. When $\eta > 0$, the analysis is less straightforward. To the authors' knowledge, it has not been determined in the literature whether or not the problem is eliminated in

this setting, and more generally whether or not the LTM is convergent. Even assuming it is convergent, the computational complexity of the method as $\eta \to 0^+$ is not obvious. This question merits further investigation, but our preliminary experiments indicate that the computational effort required to achieve a given accuracy is not uniform in $\eta$, and varies significantly with the choice of $\omega$.

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
