# Peer review of "Automatic, high-order, and adaptive algorithms for Brillouin zone integration"

_SciPost Physics, doi:SciPost Phys. 15, 062 (2023)_

## Round 1 · Referee Report · Stepan Tsirkin (Referee 1) · 2023-2-7

Strengths

1) The manuscript deals with a problem of current and broad interest.

2) The Authors develop the ``Iterative adaptive iteration'' (IAI) method and show its advantages over other methods.

3) This can help in developing efficient computational software.

4) the manuscript is clearly written and easy to follow.

Weaknesses

1) The developed IAI method is compared to a "poor man's implementation" of the good old PTR method (see report). This does not allow to evaluate the actual advantages of the new method.

2) Missing demonstrations of which features of the algorithm actually give the benefit.

Report

Although numerical integration in general is a very old problem of computational science, modern requirements on accuracy require to search for methods that are tailored to specific problems. The manuscript by Kaye et al thoroughly investigates the question of finding the most efficient method for evaluation of Brillouin zone integrals, typically arising in different problems of computational solid state physics. They develop the ``Iterative adaptive iteration'' (IAI) method and show its advantages over other methods. This is a valuable work that can help in developing efficient computational software. Also, the manuscript is clearly written and easy to follow.

However, to make the manuscript even more useful for future developments, I would like to ask the Authors to present some more analysis, in order to demonstrate which features of their Algorithms give a high improvement in practice, and which are of minor importance. Let me explain what I mean.

1) The authors start with the periodic trapezoidal rule and Algorithm 1, where on each step the number of points is incremented by a fixed number $\Delta N$, uniformly over the whole range of integration. Next, Algorithm 2 is introduced, which uses Gauss quadrature and on each step the range is split into two, and the Algorithm is recursively applied to each parts. First, I wonder why the performance of these Algorithms is not compared directly for model examples neither in 1D (like in fig 3) nor in 2D (fig 4).

2) Note that Algorithm 2 differs from Algorithm 1 by two improvements'': gauss quadrature and a tree-based refinement. (I putimprovements'' in quotes because their advantages are not demonstrated separately, although I believe that they both do give some advantage). Say, if one uses Algorithm ``2b'' with tree-based refinement, but instead of p-node Gauss quadrature a p-node (non-periodic) trapezoidal rule is used on each panel. How much worse will it be than Algorithm 2? Could you add a comparison for the 1D case (like in fig 2). How about increasing the number of nodes $p$ (upto thousands of nodes) for the whole range of integration, will it converge much better than Algorithm 1?

3) In Algorithm 1 you start with a number of points $N_{\rm trial}$, and on each step the number of iterations is increased by a fixed value $\Delta N$ (arithmetic progression). Could it be better to use a geometric progression instead, i.e. multiplying the number of points by a fixed factor $\alpha$ ($\alpha$ may also be non-integer, just $\alpha N_{\rm test}$ should be rounded at each step). Assume that the Algorithm converges to the required accuracy at some big value $N_{\rm max}\gg N_{\rm trial}, \Delta N$. Then the total number of points calculated before reaching the convergence will be $\propto N_{\rm max}^2$ in case of arithmetic progression, and only $\propto N_{\rm max}$ in case of geometric progression. Moreover, in case of geometric progression and integer $\alpha$ one can re-use the integrand calculated at all the previous points. Could this be used to improve the speed of evaluation with PTR method?

4) The authors emphasize that "when using the PTR to calculate (3) at more than one frequency $\omega$, $H(k)$ should be precomputed for all PTR grid points and stored". In turn, this causes a limitation in terms of required memory (either limited RAM, or slow disk storage) . But is it really necessary to store all k-points? Anyway, for each frequency, the evaluation will go through the same series of numbers of points $N_{test}$, until the convergence will is reached for the particular $\omega_j$. Can one modify Algorithm 1 in the following way. First, one initializes the the "convergence achieved" array $C_j=\texttt{ False}, \quad j=1,\ldots,n_\omega $. Then, doing for each $N_{\rm test}$ one does something similar to the algorithm described in the attached file

(see attached file)

And so on, increasing $N_{\rm test}$ until all $C_j$ become $\tt{True}$. This way one does not have to store any k-resolved quantity.

Of course, storage of all k-point allows to reuse the $H(k)$ evaluated at smaller $N_{\rm test}$. But anyway, the re-usage ratio will not be high, so it does not make an order of magnitude difference.

5) In appendix C the Authors mention that simple Fourier transform is faster than the zero-padded FFT when $M$ is not too large, but N is large. In my experience zero-padded FFT still can be faster for $N\sim 10 \times M$, but anyway zero-padded FFT has a huge disadvantage of the need to store all k-points in memory, which may become a problem.

However, for the case $N\gg M$ there is a solution called "Pruned FFT" (see \url{https://www.fftw.org/pruned.html} and references therein, and also J. Markel, "FFT pruning," in IEEE Transactions on Audio and Electroacoustics, vol. 19, no. 4, pp. 305-311, December 1971, doi: 10.1109/TAU.1971.1162205 ). This method was successfully employed in the WannierBerri code (S.S.Tsirkin npj Computational Materials 7, 33 (2021) ) giving a huge boost in computational time. Also, prunned FFT avoids the need to store huge interpolation grids in memory. In turn, pruned FFT cannot be used for schemes with Gauss quadratures.

Further the Authors write that "there is no straightforward method to restrict a zero-padded FFT to the IBZ". In contrast, the WannierBerri paper gives a recipe to efficiently employ symmetries in conjunction with prunned FFT (see sub section "Symmetries"), which was successfully implemented in the code.

6) In section VI, why is TAI method not used? Could you add the Wall time for TAI to fig 5 and 7, how would it compare?

7) In figures 5 and 7 , why is PTR much faster than IAI for high values of eta? Is it because it actually requres less k-points to evaluate, or is it because of some specific tricks used for faster evaluation of those k-points? I am curious to see how those methods compare in terms of needed k-points to evaluate?

8) Colormap figures (Fig 1 right, and two left panels of Fig 2) would benefit from adding subpanels with a zoom of a small region near "red line". This would improve visibility of the details, especially when printed on paper. One can use a quarter of the original panel to place a subpanel.

To summarize, I think that the manuscript by Kaye et al presents a valuable work, giving interesting solutions to a modern problem of computational materials science. However, when presenting the power of the newly derived method, I ask the Authors to present a "fair competition", i.e. to compare the IAI method with a "good implementation of PTR", which includes pruned FFT, no storage of full grid, symmetries and also geometric progression of grid sizes. How will this modify Figs 5 and 7? From the scaling relations, it looks like IAI should win in the small eta limit, but when will the lines actually cross?

Requested changes

1) compare IAI with a "good implementation of PTR", which includes pruned FFT, no storage of full grid, symmetries and also geometric progression of grid sizes. Show, ho this will this modify Figs 5 and 7.

2) Compare performance of Algorithm 2 with Algorithm 1 on a 1D example. Also add comparison with "intermediate algorithms" (see point 2 in report)

3) add TAI method to Figs 5 and 7

4) add comparison of the number of evaluated points using PTR, IAI and TAI for the example of SrVO3

Attachment

  • validity: high
  • significance: high
  • originality: high
  • clarity: top
  • formatting: excellent
  • grammar: perfect

Author:  Jason Kaye  on 2023-02-14  [id 3352]

(in reply to Report 1 by Stepan Tsirkin on 2023-02-07)
Category:
remark
answer to question
reply to objection

We thank the referee for a thoughtful and helpful report. Below, we give a detailed point by point response to the referee's comments, and also indicate our intended modifications to the manuscript. In particular, although the referee makes several interesting and useful points on the PTR which we will certainly mention in our revised manuscript, we believe that we have presented a highly efficient implementation of the PTR scheme and have thus provided a fair comparison with the IAI proposal.

(1) For 1D and 2D model functions, we feel that the benefit of adaptive integration over equispaced integration for functions with localized features is conceptually clear, and standard material. Since the degree of advantage is example-specific, we feel that additional comparisons in model cases would distract from the main points of the manuscript while increasing its length unnecessarily. An exception is Fig. 4, which, using a simple model, demonstrates a central and non-trivial conceptual point of the manuscript: that for functions with a specific structure often encountered in BZ integration, IAI is asymptotically superior to TAI. The PTR and IAI are compared for a real example in Figs. 5 and 7.

However, we will certainly add a remark comparing the number of integrand evaluations required by the PTR and adaptive integration to achieve a given accuracy, for our 1D model problem with a small value of eta.

(2) Algorithm 2 relies on a fixed 1D quadrature rule for smooth functions on an interval. The referee proposes to replace the Gauss rule (the optimal rule for non-periodic functions on an interval, exponentially convergent in p as recalled by Thm. 2) by a p-node nonperiodic trapezoidal rule (which is inferior, being merely $O(1/p^2)$ convergent). The poor accuracy of the latter scheme on each panel would result in a very large number of panels needed by Alg. 2 to achieve a small error tolerance. This would not be a fair comparison against the PTR applied globally to the periodic integrand, which in contrast to the above second-order accurate scheme, is exponentially convergent for analytic functions.

Our manuscript focuses only on high-order accurate methods, a necessary ingredient (from the point of view of efficiency) in algorithms which allow the user to specify a possibly small error tolerance. We will include a remark which explains this.

The other suggestion is to consider a global Gauss quadrature rule rather than the PTR. This would be inferior to the PTR, because the Gauss rule does not exploit the known global periodicity, whereas the PTR does. While both are exponentially convergent schemes for analytic functions, the Gauss rule is known to be the gold standard for an interval without a periodicity assumption, whereas the PTR is the gold standard for a periodic interval. This is standard material in numerical analysis, covered in Refs. [26,28,39]. Indeed, it can be shown that the convergence rate of the PTR is a factor $\pi/2$ faster than that of Gauss quadrature for a periodic function that is known only to be analytic in a given strip $|\text{Im}\, z| < a$. See, e.g., "New quadrature formulas from conformal maps," N. Hale and L. N. Trefethen, SIAM J. Numer. Anal. 46(2), 930--948 (2008). Thus there would be no point in testing the Gauss rule in the global periodic setting in which it is provably inferior.

We will make a remark addressing this point, and include a reference.

(3) The referee makes the interesting observation that, if one were starting from an $N_{trial}$ and $\Delta N$ that were smaller by a large factor than the $N_{max}$ at which the desired accuracy is achieved, then an arithmetic progression would require $O(N_{max}^2)$ evaluations and would therefore be inefficient. However, this is not the situation that we describe. Rather, we choose $\Delta N$ via a rigorous estimate that scales as $1/\eta$, as detailed in App. A, so that only $O(1)$ adaptation steps are taken. Thus we stand by the decision to use an arithmetic rather than a geometric progression of grid sizes in our efficient implementation of the automatic PTR.

To elaborate, since the PTR converges exponentially, once one is in the convergence regime, adding a fixed number $\Delta N$ of points decreases the error by approximately a fixed factor. Thus, a good strategy is to estimate a number $N_{trial}$ sufficient to be in the convergence regime (but not overkill), and then to repeatedly add $\Delta N$ points in order to drive the error down exponentially below the tolerance. As a consequence of the exponential convergence, this is expected to take at most a fixed, constant number of steps, giving a total number $O(N_{max})$ of points calculated, not $O(N_{max}^2)$. We will add a remark in the manuscript clarifying how $N_{trial}$ is chosen.

If a geometric progression of grid sizes were used, this would in certain cases result in significant over-convergence, and therefore a significantly larger cost. For example, if one chose $\alpha = 2$, one would often encounter situations like the following: $N = 1100$ is sufficient for convergence below $\epsilon$, but one has $N_{trial} = 1000$ and $N_{test} = 1000 \times 2 = 2000$. In this case, one does $2000^3 / 1100^3 \approx 2^3 = 8$ times more work than is needed. Using an arithmetic progression of grid sizes, with spacing chosen using $\eta$, avoids this common scenario.

The benefit of reusing previously computed values is only valid for discrete choices of $\alpha$, the smallest of which is $\alpha = 2$. Therefore, in 3D, the final calculation will be so much more expensive than the previous ones that the savings gained by reusing the computed values will be insignificant.

(4) This is a very interesting suggestion and has useful advantages: it avoids storing $H(k)$ (which also increases the flops-to-DRAM-access ratio), and it avoids using over-resolved grids for frequencies which may only require a coarse grid. However, this algorithm has the disadvantage that it requires specifying all frequency evaluation points in advance. In our case, we use adaptive interpolation in the frequency domain, and therefore our frequency evaluation points are determined on the fly.

If one wishes to evaluate the Green's function, for example, at a pre-specified frequency grid, the algorithm proposed by the referee may be preferable. However, if one wishes to compute the Green's function to a user-specified accuracy $\epsilon$, as is the goal in our manuscript, the frequency grid must resolve features at $O(\eta)$ scale which are, in general, at unknown locations. Therefore, an a priori-specified frequency grid would require $O(1/\eta)$ points. We avoid this using an adaptive algorithm, described in Sec. V, which determines an efficient frequency grid of $O(\log(1/\eta))$ points on the fly, and is therefore not compatible with the algorithm proposed by the referee. We do not feel the possible advantage gained by switching to this algorithm is sufficient to justify replacing our $O(\log(1/\eta))$ scaling with $O(1/\eta)$ scaling.

We do not believe that our goal of computing the Green's function automatically to a user-specified tolerance is contrived. Although in some cases one only wants, for example, to roughly plot the density of states, in other cases the calculation of the Green's function appears in the inner loop of a complicated procedure (as in dynamical mean-field theory). In that case it may be required to perform the calculation automatically within a required error tolerance. We therefore feel that both algorithms have their place, and will make a remark explaining this in our revised manuscript.

(5) The suggestion of using a pruned FFT algorithm is interesting, but we do not feel that it is likely to give a significant advantage over the method we have proposed, for the following reasons.

Let $n_R$ denote the number of Fourier coefficients of $H(k)$ per dimension, with $N_R = n_R^3$, and let $n_k$ denote the number of $k$-evaluation points per dimension, with $N_k = n_k^3$. Although it is stated in the given reference (Tsirkin, npj Comput. Mater. 7, 33 (2021)) that direct evaluation of the Fourier series has a cost scaling as $O(N_R N_k)$, this is only the case if one evaluates the full 3D Fourier series at each $k$-point. However, since the $k$-points fall on a Cartesian grid, this is unnecessary and inefficient. Rather, as described in Sec IV.B (subsec. "Rapid evaluation of $H(k)$ in iterated adaptive integration") and in Appendix C, one should split the Fourier series dimension-by-dimension, so that the dominant cost is that of evaluating a one-dimensional Fourier series of $n_R$ terms at each of $N_k$ points. Each such Fourier series evaluation is extremely efficient when the geometric series property (addition formula) is exploited, requiring a single complex exponential evaluation, $2n_R$ multiplications, and $n_R$ additions. The computational complexity is therefore $O(N_k n_R)$, not $O(N_k N_R)$. This method also allows us to take full advantage of all crystal symmetries, as explained in Appendix C.

The proposed pruned FFT algorithm has complexity $O(N_k \log n_R)$. While this is asymptotically less than $O(N_k n_R)$, it is difficult to state a priori which method will be faster in practice, since $n_R$ is small and the direct evaluation algorithm has an extremely small prefactor. Furthermore, it appears that the pruned FFT algorithm does not necessarily take advantage of all symmetries for typical grid densities, which could be a significant disadvantage in practice.

Therefore, while a direct comparison of these approaches would be an interesting topic of future research, we feel that we have implemented a competitive and likely optimal Fourier series evaluation algorithm, and do not wish to distract from the main points of the paper by including this. We will make a remark in the text referring to the pruned FFT algorithm, and a summary of the points made above.

(6) The TAI method was eliminated in Section IV on a variety of grounds, and is highly unlikely to be competitive with IAI in practice. Indeed, Fig. 4 shows for a model problem that TAI requires more integrand evaluations than IAI, even for large $\eta$, and each integrand evaluation is unlikely to be more efficient in TAI than in IAI. Furthermore, integration on the irreducible Brillouin zone would require designing a TAI scheme with tetrahedral cells, and high-order quadratures on tetrahedra. This would be a very significant undertaking, and makes it difficult to include TAI, which is unlikely to be competitive in any case, in the comparisons in Sec. VI. Neither would it be fair to run the cube TAI on the full Brillouin zone and compare with IAI and PTR on the irreducible Brillouin zone.

(7) The PTR is expected to be faster than IAI for large values of $\eta$ for a few reasons. First, for large $\eta$, one does not need much adaptivity. In this case, IAI will produce panels of roughly equal size; if they were equal, this would yield a method of order $2p$ (for us, $p = 4$), as opposed to the spectrally convergent PTR. For smooth, periodic functions which do not require $k$-space adaptivity, the PTR is an optimal, highly efficient method.

Second, if one evaluates at multiple frequency points, the PTR can reuse previously stored values of $H(k)$, whereas for IAI $H(k)$ needs to be evaluated on the fly. As we state in our manuscript, this is a very significant advantage. To make a comparison which is sufficiently generous to the PTR, the cost of evaluating $H(k)$ was not even included in the PTR timings in Fig. 5, whereas obviously for IAI this evaluation does contribute to the timings (we will state this in the revised manuscript).

Lastly, the IAI method comes with some overhead which the much simpler PTR does not. The PTR is significantly easier to optimize at a low level, and we put considerable work into optimizing our implementation of the PTR in order to provide a fair comparison.

The number of function evaluations used by IAI and the PTR is not a fair metric for comparison of performance, because the cost per function evaluation is significantly different for the two methods. The most significant reason is that in IAI, $H(k)$ must be evaluated on the fly. For this reason, we have opted to only show comparisons using wall clock timings.

(8) This is a good idea, and we will try to add a corner zoom to these figures.

We again thank the referee for these thoughtful comments, which will lead to significant improvements in the clarity of the manuscript. We believe our responses should alleviate any concerns that we have not presented a high-performance implementation of the PTR.

Stepan Tsirkin  on 2023-03-08  [id 3447]

(in reply to Jason Kaye on 2023-02-14 [id 3352])

I would like to thank the Authors for a detailed reply. This is very convincing. I have learned a lot both from the paper and from the reply.

I will be happy to recommend this manuscript for publication in SciPost after the Authors implement the clarifications proposed in the reply.

---

## Round 1 · Referee Report · Anonymous (Referee 2) · 2023-2-17

Strengths

1 – The manuscript discusses an important issue, which needs to be tackled in many parts of modern (possibly ab initio based) descriptions of correlation phenomena in solid state systems.

2 - The manuscript provides a very detailed overview of conventional and adaptive Brillouin zone integration techniques, with the potential to become a great reference for the topic.

3 - The presented iterated adaptive integration (IAI) scheme with user-defined precision seems to outperform the tree-based adaptive and more conventional trapezoidal integration scheme, while being rather simple at the same time.

4 – All important dimensions are simultaneously discussed.

Weaknesses

1 – Although the showcase example problem defined in Eq. (3) with a local, i.e. k-independent, retarded self-energy is a very important example for the, e.g., dynamical mean field and coherent potential theory communities, it does not touch upon other wide-spread non-local self-energies as, e.g., appearing in GW or Eliashberg calculations.

2 – Rather small single-particle Hamiltonians are used (in terms of their orbital basis).

3 – For the “Frequency Domain Adaptivity” it is assumed that the retarded self-energy can be evaluated at arbitrary frequencies.

4 – The manuscript is partially hard to read as a result of the strict mathematical tone.

Report

The manuscript is well and clearly written (modulo the strict mathematical tone) and fulfills all general acceptance criteria defined by Scipost. Regarding Scipost’s expected requirements on the groundbreaking / breakthrough / new pathway / synergy nature of the manuscript required for acceptance, my personal impression is that the manuscript does not check one of them specifically, but rather checks all of them to a certain degree: The problem of a precise and reproducible Brillouinzone integration up to user-defined precision is a previously-defined and long-standing issue of the solid state theory community (and especially of that part of the community that concentrates on low-energy / many-body problems). The synergy between algorithmic thinking and mathematical handling in combination with the concept of adaptive grids might also be seen as a combination of fields, which allows for the derivation of an impressively performing algorithm.

I, however, also see a short coming in the manuscript at hand resulting from the simple structure of the assumed self-energy. As already mentioned above as a weakness, there are plenty examples in which the self-energy carries its own non-locality. As the adaptive integration strongly relies on the ability to evaluate H(k) at arbitrary momenta, which is given here due to its tight-binding like structure, the efficiency of the presented IAI might be strongly limited as soon as the self-energy evaluation at a specific k requires significant computational resources. A similar issues arises from the frequency domain adaptivity, but with respect to w.

I nevertheless see the possible impact the presented algorithm can have for the (equally plenty) situations with strictly local self-energies.

To conclude, I would be in favor of publication in Scipost if the authors would (A) add corresponding discussions to their manuscript and would (B) provide possible strategies to tackle non-local self-energies.

Requested changes

see above

---

## Round 2 · Referee Report · Stepan Tsirkin (Referee 1) · 2023-4-4

Report

The autoors have adequately responded to the previous comments and made corresponding changes in the manuscript. I recommend the paper for publication

---

## Round 2 · Referee Report · Anonymous (Referee 2) · 2023-4-12

Report

I fully agree with my colleague and recommend the paper for publication.

---

## Round 2 · Author Response

We again thank the referees for their thorough comments on our manuscript. We have made a variety of modifications in response to the first referee report, most of which were mentioned in our previous response. We provide a detailed list of all changes to the manuscript below.

In addition, we wish to provide a brief response to the main point raised by the second referee concerning $k$-dependent self-energies. As long as the self-energy can be evaluated rapidly on the fly, the generalization to this case is straightforward. There are then two points to make about this issue. First, although the self-energy may be expensive to evaluate in its given form, many standard tools exist to replace a representation of a function which is expensive to evaluate by another which can be made essentially as fast to evaluate as needed (to take an extreme case, for example, one could pre-tabulate a function on a very dense grid, and use a local linear polynomial interpolant for rapid evaluation). This preparation of the self-energy is a precomputation. Using such a scheme, the efficiency of integration can be made as good as the underlying structure of the self-energy allows. Second, the most natural fast-to-evaluate representation is problem-dependent. For example, the most efficient representation might be on an adaptive grid (like the adaptive Chebyshev interpolant described in the manuscript); it might be a Fourier series in $k$ with $\omega$-dependent coefficients represented by a polynomial interpolant; or it might be a local piecewise-polynomial interpolant on a uniform grid. We have added a remark to the conclusion which summarizes these points.

We welcome feedback from the referees on whether our modifications address their concerns satisfactorily.

---

## Round 2 · List of Changes

Changes in response to first referee report, using the referee's numbering:

(1) In Section IIIB we have given a comparison between the PTR and adaptive integration for the 1D example described in Section II. (2) We have added a sentence in the introduction explaining why our manuscript focuses on high-order methods. (2) We have added a sentence at the end of Section IIIA noting that the PTR is superior to Gauss quadrature for periodic function, with a reference. (3) In Section IIB we have specified the initial value of $N_{\text{trial}}$ that we use. (4) We have added a remark in Section IIB, as well as an Appendix describing the suggested algorithm. We have also modified Algorithm 1 and the surrounding discussion to make it clear that it is not necessary to specify the list of evaluation frequencies in advance. (5) We have modified Appendix C. In particular, we added a discussion on the pruned FFT, with references, and have removed the statement that there is no efficient algorithm to restrict a zero-padded FFT to the irreducible Brillouin zone. (7) We have added a sentence in the caption to Fig. 5 explaining that the time to evaluate $H(k)$ is not included in the timings reported in the figure. (8) We have modified Fig. 2 to improve the contrast and clarity of the grid points.

Changes in response to the second referee report:

We have added a short remark in the introduction, and a longer explanation in the conclusion, addressing the generalization to $k$-dependent self-energies.

Other changes

  • We have corrected typos in the definition of the function used in Fig. 2, and in Eq. 7.
  • We have made a few minor notational changes and remarks.

---

## Editorial Decision

published